# The transcriptional response of cortical neurons to concussion reveals divergent fates after injury

Mor R. Alkaslasi [1,2], Eliza Y. H. Lloyd[1], Austin S. Gable[1], Hanna Silberberg[1], Hector E. Yarur[3], Valerie S. Tsai[3], Mira Sohn[4], Gennady Margolin[4], Hugo A. Tejeda [3] & Claire E. Le Pichon [1] ✉

Traumatic brain injury (TBI) is a risk factor for neurodegeneration, however little is known about how this kind of injury alters neuron subtypes. In this study, we follow neuronal populations over time after a single mild TBI (mTBI) to assess long ranging consequences of injury at the level of single, transcriptionally defined neuronal classes. We find that the stress-responsive Activating Transcription Factor 3 (ATF3) defines a population of cortical neurons after mTBI. Using an inducible reporter linked to ATF3, we genetically mark these damaged cells to track them over time. We find that a population in layer V undergoes cell death acutely after injury, while another in layer II/III survives long term and remains electrically active. To investigate the mechanism controlling layer V neuron death, we genetically silenced candidate stress response pathways. We found that the axon injury responsive dual leucine zipper kinase (DLK) is required for the layer V neuron death. This work provides a rationale for targeting the DLK signaling pathway as a therapeutic intervention for traumatic brain injury. Beyond this, our approach to track neurons after a mild, subclinical injury can inform our understanding of neuronal susceptibility to repeated impacts.

Close to half the population is expected to experience a mild traumatic brain injury (mTBI) at some point in their life[1]. A common form of mTBI is concussion, a brain injury caused by mechanical force and resulting in temporary neurological dysfunction. Although most people seemingly recover, for some the impact can lead to long-term damage. There is increasing evidence that repeated mTBI can cause chronic traumatic encephalopathy (CTE)[2] and is a potential risk factor for other neurodegenerative disorders[3–5]. The primary insult of mTBI triggers a cascade of damage termed 'secondary injury' that involves multiple brain cell types and unfolds during the days and weeks following the impact[1]. Even if symptoms during this time can be relatively mild, it is during this chronic phase that neurons are thought to become more

vulnerable to repeated injuries. Despite this, the typical treatment for mild TBI is limited to pain management and rest. It is therefore likely that neuroprotective treatments would be beneficial to stave off risk of permanent damage.

To develop effective treatments, we must first better understand the pathways initiated in particular neurons. Many studies to date have lacked the resolution to discern cell type-specific responses[6–8]. More recent work has examined the effect of injury on particular neuron types[9,10]. We reasoned that a detailed look at a model of single mild TBI would provide important insight into the nature and extent of neuronal injury immediately following a concussion. We previously generated a mouse line to track neurons that are transcriptionally

[1]Unit on the Development of Neurodegeneration, Eunice Kennedy Shriver National Institute of Child Health and Human Development, National Institutes of Health, Bethesda, MD, USA. [2]Department of Neuroscience, Brown University, Providence, RI, USA. [3]Unit on Neuromodulation and Synaptic Integration, National Institute of Mental Health, National Institutes of Health, Bethesda, MD, USA. [4]Bioinformatics and Scientific Programming Core, Eunice Kennedy Shriver National Institute of Child Health and Human Development, National Institutes of Health, Bethesda, MD, USA. ✉e-mail: claire.lepichon@nih.gov

responsive to peripheral nerve injury[11] and wondered if it could be used to investigate mTBI.

Peripheral nerve injuries cause a transcriptional response in sensory neurons that is regulated by *Atf3* (activating transcription factor 3) and is essential for functional recovery[11,12]. In sensory neurons, *Atf3* is responsible for upregulating select regeneration-associated genes while repressing many other genes during the recovery process. We wondered whether a similar transcriptional response might occur in the brain after mTBI. Previous studies have observed neuronal *Atf3* activation following injury to the central nervous system. *Atf3* was reported to be activated in cortical neurons following TBI and in corticospinal neurons by axon transection depending on the proximity of the injury to the soma[13,14]. Studies in *Atf3*-deficient mice found worse outcomes following TBI and ischemia[15,16], suggesting a protective role for *Atf3*, but did not distinguish between neuronal and glial activation of Atf3. We therefore hypothesized Atf3 would be activated in neurons after mTBI, but wondered how these neurons would compare to peripheral neurons in their ability to exhibit plasticity and recover.

Using genetic reporter mice, single nucleus RNA-sequencing, and slice electrophysiology, we performed a detailed characterization of the neurons that transcriptionally activate *Atf3* after mTBI. We demonstrate that several subclasses of cortical neurons engage the *Atf3* response, but that these undergo divergent fates (death vs survival) that are linked to their identities. We probe the role of multiple candidate pathways for their contribution to cortical neuron death after mTBI and find that dual leucine zipper kinase (DLK), an upstream regulator of *Atf3*, drives neuron death in layer V, highlighting it as a potential therapeutic target for mTBI. These results underscore a differential vulnerability of cortical neurons to mTBI and emphasize the importance of studying injury-induced pathology at the level of individual neuronal subtypes.

## Results

### Characterizing neurodegeneration in a closed skull model of mild TBI

To study mild TBI, we characterized a unilateral closed-skull injury model[17] that provides a clinically relevant view of concussion injury and allowed us to accurately dissect the resulting pathological cascade. We used a controlled cortical impact injury wherein an impact was delivered directly to the surface of the skull at a specified depth and velocity. The impact was provided by a 3 mm diameter tip positioned over the mouse's left sensorimotor cortex (Fig. 1a). Following this injury, mice presented with no overt long-term symptomology and no tissue loss, but did exhibit reproducible cortical astrogliosis in an area ~2 mm in diameter and confined to the ipsilateral cortex (Fig. 1b). This model resulted in a loss of righting reflex concordant with mild TBI (righting time < 15 min[18,19], Fig. 1c), as well as a small yet consistent increase in the serum biomarker of neuron degeneration, neurofilament light[20] (NfL, average 3-fold higher than baseline between 1 and 14 dpi, Fig. 1d).

We started by examining pathology in the Thy1-YFP-H mouse, which sparsely expresses a fluorescent protein in layer V cortical neurons, highlighting their morphology[21,22]. We histologically confirmed cortical neurodegeneration in these mice (Fig. 1e). At 7 days post injury (dpi), we observed hallmarks of degenerating dendrites, cell bodies, and axons specifically in the ipsilateral cortex but not the side contralateral to injury (Fig. 1e–k and Supplementary Fig. 1a). Cortical dendrite fragmentation in the region above layer V was quantified using a degeneration index calculation[23] and revealed significant degeneration of YFP+ dendrites[24] only in the ipsilateral cortex when compared to the contralateral side or the ipsilateral cortex of sham injury controls (Fig. 1f, i). Below layer V there was a significant increase in YFP-positive structures that did not correspond to cell bodies but rather to pathological enlargements of axons (area = 10–250 $\mu m^2$, mean = 20 $\mu m^2$, Fig. 1h, j). Some of these axonal swellings were of comparable size to cell bodies but none contained DAPI-positive nuclei (Supplementary Fig. 1b). The swellings likely correspond to disruption in axon transport leading to organellar and protein accumulations, also called diffuse axonal injury[9,25–27]. We also observed beading of axons (fragments < 10 $\mu m^2$) representing axon degeneration. We found that both axon beading and swelling were increased only in the ipsilateral cortex at 7 dpi (Fig. 1h, j).

Because the Thy1 reporter is stochastically expressed, and because we had specifically observed inflammatory, dendritic, and axonal pathology only on the side ipsilateral to injury, to quantify any potential cell loss, we compared the number of YFP+ neurons in the cortex ipsilateral to the injury, and normalized them to the contralateral cortex of each section. We measured a 15.3% ± 1.8 loss of cell bodies at 7 dpi, and 26.3% ± 7.9 loss at 14 dpi (Fig. 1e, k). Thus, we find that a single unilateral closed-head impact over the sensorimotor cortex reproducibly leads to degeneration of layer V projection neurons ipsilateral to the injury and across neuronal compartments.

### mTBI produces an Atf3 response in a subset of cortical neurons

Transcription factors play major roles in neuronal responses to injury, and their activation can determine whether a neuron degenerates or regenerates. The transcription factor ATF3, in particular, is a master regulator of the transcriptional response to neuronal injury and is responsible for driving a transcriptional shift toward an injured cell state[11,12]. We looked for ATF3 expression in brains following mTBI and observed that at 7 dpi, ATF3 immunolabeling localized specifically to the injured side of the cortex (Fig. 2a). We found 10% ± 2.3 of YFP-expressing neurons expressing ATF3 at this timepoint[13] (Fig. 2b). Some ATF3-positive cells were also present in layer II/III (Fig. 2a).

Our initial attempts to identify the ATF3-positive neurons in layer V using markers of projection neurons such as CTIP2[28] were unsuccessful. We observed no double-labeled cells upon staining for ATF3 and CTIP2 (Fig. 2d–f). We reasoned that ATF3 might be repressing marker genes in the cortex after TBI as has been observed in peripheral sensory neurons after axon injury[11,12]. We therefore performed single nucleus RNA sequencing of these neurons to obtain a more comprehensive picture of their repertoire of RNA expression.

### Transcriptional profiling of cells that activate Atf3 in the injured cortex

We employed targeted snRNAseq of *Atf3*-expressing neurons using an inducible Atf3-IRES-CreER mouse line[12] crossed to the INTACT nuclear envelope protein reporter[29]. The resulting animals express GFP tethered to the nuclear envelope in cells that are expressing Atf3 at the time of tamoxifen treatment. Tamoxifen was administered at 4 and 5 dpi to induce expression of the nuclear GFP reporter and cortical tissue at the injury site was collected at 7 dpi, enabling the isolation of nuclei for single nucleus sequencing from cells expressing *Atf3* during this acute phase (Fig. 2c).

We sequenced 8065 GFP+ nuclei and found a significant number of microglia (4274, 53.0%), excitatory (2057, 25.5%) and inhibitory (744, 9.2%) neurons, and small populations of astrocytes (107, 1.3%), oligodendrocytes (75, 0.9%), and other cells (808, 10%) (Supplementary Fig. 2a–e). In this study we focus on the role of ATF3 as an injury marker in neurons, but we note its role in microglial function as an interesting avenue for future investigation[30,31].

### Specific subclasses of excitatory and inhibitory cortical neurons activate Atf3-associated injury pathways

We mapped the neuronal nuclei from this experiment onto a highly annotated mouse motor cortex reference atlas[32] to assign cellular subclasses based on the nuclear transcriptome. Of the 2801 neuronal nuclei sequenced using the Atf3-CreER approach, we identified excitatory neurons across cortical layers, parvalbumin (Pvalb) and somatostatin (Sst) interneurons, and small numbers of Lamp5, Vip, and Sncg

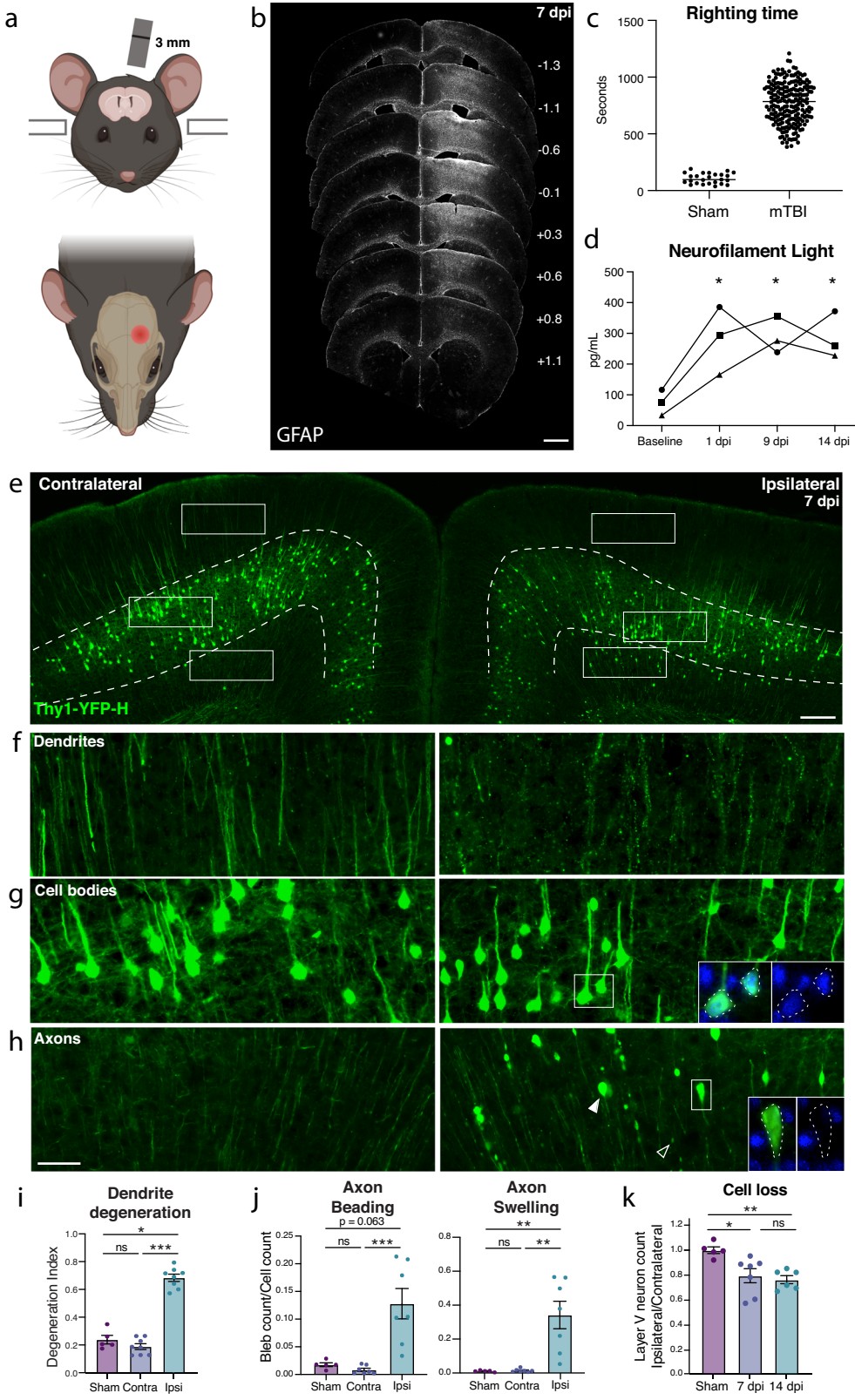

interneurons (Fig. 2d, f, Supplementary Fig. 2a, b, and Supplementary Data 1). Using multiplex in situ hybridization with markers from the data, we validated the presence of Gfp+ excitatory and inhibitory neurons (Supplementary Fig. 3a–c). Interestingly, all interneuron subclasses expressed the inhibitory neuron marker *Gad2*, but lacked their subclass markers, including *Pvalb* or *Sst* (Fig. 2f). Similarly, neurons assigned to excitatory cortical layer identities lacked the typical

expression of *Slc17a7* (VGLUT1), but expressed some layer-specific markers, such as *Cux2*, *Rorb*, and *Foxp2* (Fig. 2f). Thus, in the cortex– similar to the peripheral sensory system[11,12]–Atf3 expression leads to the downregulation of multiple marker genes.

In our experiment, we captured neurons that had expressed *Atf3* at 4 and 5 dpi, and sequenced them at 7 dpi. Some of these still expressed *Atf3*, but many only had low or undetectable *Atf3* expression

**Fig. 1 | Closed-skull mTBI induces layer V neuron degeneration and death.**
**a** Schematic of injury model/location. Coronal view (top) shows tip positioned over left cortex. Horizontal view (bottom) shows impact site relative to bregma.
**b** Example of extent of injury, representative of >50 samples in which immunostaining for neuroinflammation was performed. GFAP staining is shown across sections from anterior to posterior. Approximate mm from Bregma on the anterior-posterior axis shown on the right. **c** Quantification of righting times (time to wake from anesthesia) demonstrating loss of righting reflex in TBI animals consistent with mild TBI. N = 25 Sham, 218 mTBI. All wildtype mice in the study included, excluding any without accurately recorded righting times. **d** Longitudinal measurement of neurofilament light in serum of mTBI animals. Each shape and line represents measurements for one mouse and the average of two replicates per mouse for n = 3 mice. *p = 0.0421 (1 dpi), p = 0.0352 (9 dpi), and p = 0.0381 (14 dpi) by Tukey's multiple comparisons test for each timepoint compared to baseline.
**e** Low magnification image of ipsilateral and contralateral cortex in Thy1-YFP-h

mice. Layer V is outlined. High magnification images of **f** dendrites, **g** cell bodies, and **h** axons in the contralateral and ipsilateral cortices. For ipsilateral cell bodies and axons, insets show DAPI expression in cell bodies, and lack of DAPI expression in axon swellings. **i** Quantification of dendrite degeneration at 7 dpi in sham (n = 5), contralateral (n = 8), and ipsilateral (n = 8, *p = 0.0302, ***p = 0.0007 by Kruskal–Wallis test). **j** Quantifications of axon beading (fragments with area < 10 μm², ***p = 0.0009 by Kruskal–Wallis test) and axon swellings (fragments with area > 10 μm², **p = 0.007 by Kruskal–Wallis test) at 7 dpi in sham (n = 5), contralateral (n = 7), and ipsilateral (n = 7). **k** Quantification of YFP+ neurons in ipsilateral compared to contralateral cortex in sham animals and at 7 (n = 7) and 14 (n = 6) dpi compared to sham (n = 5, *p = 0.0378, **p = 0.0073 by Kruskal–Wallis test). For **i**–**k** points represent the average of 3–4 sections per mouse. Error bars represent SEM. ns not significant, by unpaired t-test. Scale bars: **b** 1 mm, **e** 200 μm, **f**–**h** (shown in **h**) 50 μm. Created in BioRender. Le Pichon, C. (2025) https://BioRender.com/y05i143.

(Fig. 2g and Supplementary Fig. 3a). In contrast, the damage-induced neuronal endopeptidase *Ecel1*, whose expression is directly downstream of *Atf3*[33], was highly expressed in both excitatory and inhibitory neurons (Fig. 2g and Supplementary Fig. 3a). We investigated the expression of a panel of injury-induced genes across neuronal subclasses and discovered that not all neuron subclasses that activated *Atf3* underwent the same subsequent transcriptional programs (Fig. 2g). For example, pro-apoptotic and endoplasmic reticulum (ER) stress genes such as *Ddit3*[34] were most highly expressed in layer V neurons and low in layer II/III (validated by in situ hybridization, Supplementary Fig. 3d), while *Atf3* and axon growth genes were most highly expressed in Pvalb and other interneuron subclasses. Thus although many neurons upregulate Atf3, their overall transcriptional changes differ according to cell type. Our data thus highlight heterogeneous transcriptional programs and fates among the Atf3-captured neurons.

### Genetic labeling of Atf3-expressing neurons highlights layer-specific vulnerability

To visualize and map the neurons that express *Atf3* after mTBI, we generated a neuron-specific *Atf3* reporter mouse (Atf3-GFP) in which GFP is permanently expressed only in neurons once *Atf3* is upregulated. This mouse results from a cross between Atf3-IRES-Cre and a Cre-dependent reporter line expressing GFP under control of the neuron-specific Snap25 promoter (Jax 021879). In control mice, sparse GFP labeling is observed in the cortex and in some hippocampal neurons, likely due to developmental Atf3 expression (Supplementary Fig. 4a). We assessed the extent of Atf3-GFP labeling in the cortex and in other brain regions, observing injury-induced GFP primarily on the ipsilateral side of the cortex, as well as in the ipsilateral anterior thalamic nuclei (Supplementary Fig. 4b). The anterior thalamic neurons project into the cortex around the site of injury, and thus their axons may be damaged in this injury model[35,36].

Because this Cre-dependent system results in permanent labeling of neurons in which *Atf3* is induced, we concluded that loss of GFP-expressing neurons together with signs of neurodegenerative pathology would indicate cell death. Longitudinal quantification up to 70 dpi of GFP-expressing neurons in the ipsilateral cortex revealed that a prominent group of layer V cortical neurons expressed Atf3-dependent GFP between 5 and 10 dpi and subsequently disappeared by 14 dpi, while a population of layer II/III neurons persisted at 70 dpi (Fig. 3a, b). The loss of layer V Atf3-GFP neurons by 14 dpi echoes the layer V neuron loss observed in the Thy1-YFP mouse (Fig. 1e, f). Parallel quantification of ATF3 protein revealed a comparative delay in GFP expression and extensive activation of ATF3 in non-neuronal cells, which was consistent with our snRNAseq data (Supplementary Fig. 5a). Interestingly, amplification of GFP signal with immunolabeling revealed that some layer II/III neurons initially exhibit lower expression of GFP (Fig. 3b).

### Layer V Atf3-expressing neurons undergo cell death and are phagocytosed following mTBI

Atf3 activation promotes axon regeneration in peripheral injuries, but here we found its expression in neurons that are unlikely to regenerate. Because GFP+ neurons at 7 dpi were present only on the ipsilateral side of the cortex and primarily in layer V (Supplementary Fig. 5c), we were able to use the stereotyped projection patterns of layer V neurons to inspect the dendrites and axons of these neurons. Similarly to the Thy1-YFP+ layer V neurons (Fig. 1e–k), we discovered that the GFP+ dendrites were severely fragmented, indicating dendrite degeneration, while GFP+ axons exhibited axonal swellings typical of diffuse axonal injury (Supplementary Fig. 5a). Morphologically unhealthy neurons that exhibited cell body vacuolization and loss of nuclear DAPI signal were also observed in layer V (Supplementary Fig. 5b, d).

Neuron death in layer V was confirmed by examining the expression of several types of apoptotic markers in Atf3-GFP tissue. The pro-apoptotic gene *Ddit3*[34] was significantly increased in GFP+ neurons compared to GFP-negative neurons either ipsilateral or contralateral to injury. In GFP+ neurons with high *Ddit3* expression, we observed high DAPI intensity reflecting chromatin condensation during apoptosis[37] (Supplementary Fig. 5d, e). These apoptotic cells had lower GFP expression and appeared morphologically misshapen (shriveled/deformed, Supplementary Fig. 5c). Additionally, the DNA damage marker phospho-γH2AX, which is phosphorylated during apoptosis[38], was elevated in GFP+ neurons at 10 dpi but not in GFP-neurons in the ipsilateral or contralateral cortex (Supplementary Fig. 5f, g). The specificity of phospho-γH2AX upregulation to GFP+ neurons and not their GFP- neighbors highlights that this mechanism of cell death is specific to neurons undergoing *Atf3*-associated injury responses at this timepoint. Thus, we conclude that layer V neurons that activate *Atf3* undergo apoptosis in the weeks following injury.

Related to this neuron death, we found that microglia exhibited increased phagocytic activity in the ipsilateral cortex and engulfed debris from dead GFP+ neurons. A significant proportion of CD68+ microglial lysosomes contained GFP+ debris (10% ± 4.2 of at 7 dpi) which increased by 14 dpi (15% ± 3.5) and coincided with the maximal loss of layer V neurons (Supplementary Fig. 5h, i). Microgliosis occurred specifically in the ipsilateral cortex where it peaked around 10 dpi and returned to baseline by 42 dpi (Supplementary Fig. 6a, b). Astrogliosis occurred in a delayed yet prolonged peak of GFAP expression which remained elevated at 70 dpi (Supplementary Fig. 6a, c).

These findings demonstrate mTBI leads to activation of *Atf3*-associated pathways in layer V cortical neurons, to their degeneration and death within two weeks after injury, and to cortical glial responses.

### Layer II/III Atf3-expressing neurons survive and remain electrophysiologically active

The analysis of Atf3-GFP neurons across cortical layers over time (Fig. 3a) revealed the loss of most GFP-positive layer V neurons, and

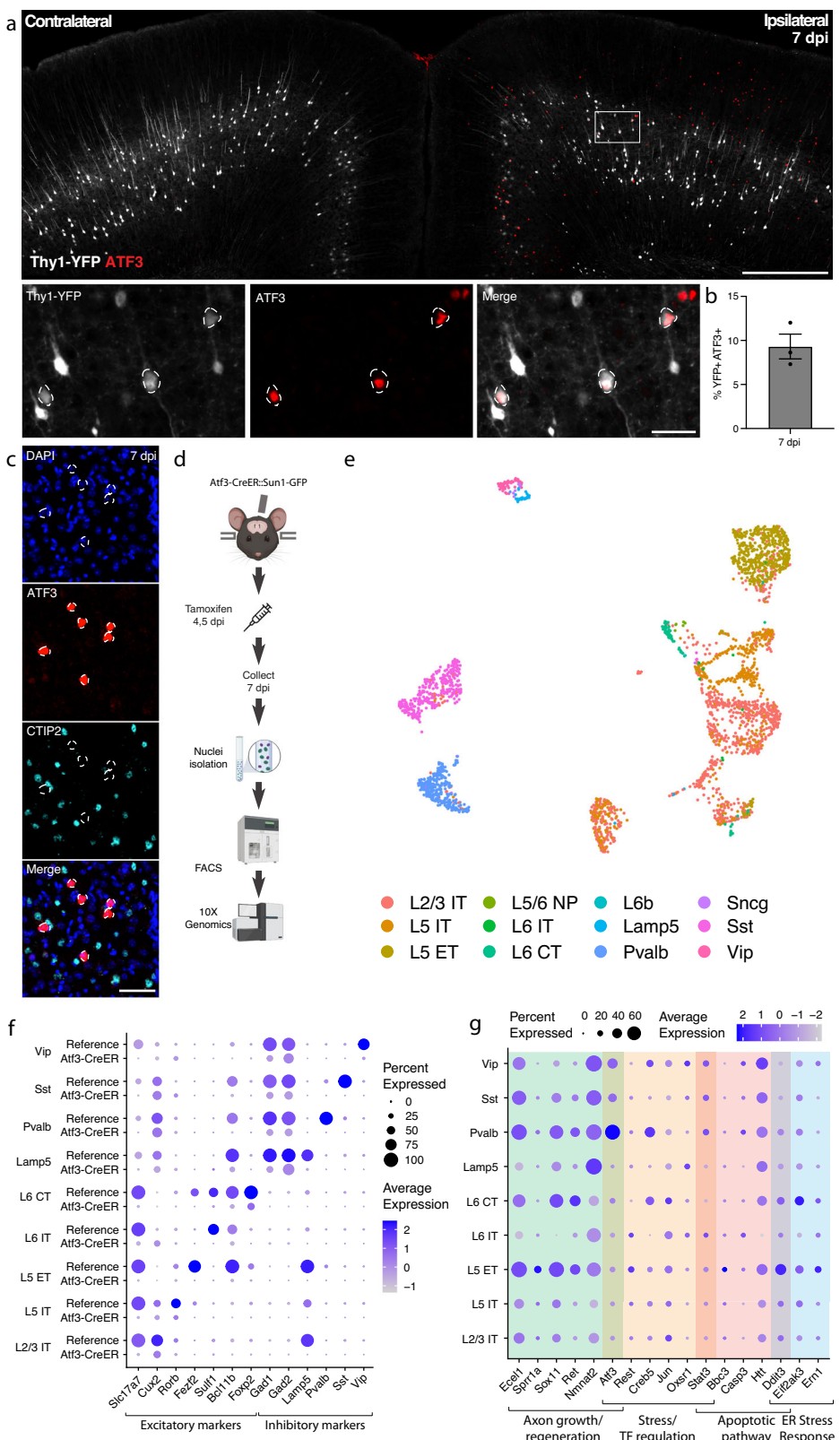

highlighted those in layer II/III as the main surviving GFP-positive population at 70 dpi. To evaluate whether this resulted from long-term survival of *Atf3*-marked neurons, we used the inducible Atf3-CreER reporter line. By injecting tamoxifen at 4 and 5 dpi, we could permanently label cells expressing *Atf3* at this time point and evaluate their localization over time (Fig. 3c). At 7 dpi, most labeled neurons were located in layer V, while those remaining at 21 dpi were primarily found in layer II/III. These layer II/III *Atf3*-marked neurons persisted until at least 42 dpi (Supplementary Fig. 7a), and likely represent a resilient population of neurons that activate this stress response pathway soon after injury and survive. Additionally, layer II/III neurons at 21 dpi appeared morphologically healthy, unlike the degenerative profiles observed in some layer V neurons at 7 dpi (Supplementary Fig. 7b, c).

**Fig. 2 | mTBI activates an Atf3-associated transcriptional response in different subclasses of cortical neurons. a** Immunostaining of ATF3 (red) in a Thy1-YFP (white) mouse showing specific expression in the ipsilateral cortex, with higher expression in layer V. Inset highlights YFP+ ATF3+ neurons. ATF3+ neurons are outlined. **b** Quantification of percent of YFP+ neurons expressing ATF3 at 7 dpi in the ipsilateral cortex. Points represent the average of 3–4 sections per animal across *n* = 3 animals. Error bars represent SEM. **c** Immunostaining showing that CTIP2 (cyan) is not expressed in layer V ATF3+ (red) nuclei. ATF3+ nuclei are outlined. **d** A schematic representation of the single nucleus RNA sequencing workflow, including unilateral closed-skull CCI, tamoxifen dosing, nuclear isolation, and FACS, following by 10X Genomics sequencing. **e** UMAP showing neurons collected

by snRNAseq of ipsilateral cortex from pooled Atf3-CreER animals, annotated by mapping to a reference atlas. For subsequent analyses, cell types with fewer than 20 nuclei are excluded. **f** Dotplot of marker genes for layer-specific excitatory neurons and subclasses of inhibitory neurons, showing downregulation of some markers in Atf3-CreER animals compared to the reference dataset. **g** Dotplot of a panel of known stress response genes involved in axon growth and regeneration, cell stress and transcription factor regulation, apoptosis, and ER stress highlighting different responses between *Atf3*-expressing neuron subclasses. Genes were selected based on altered expression compared to reference dataset. Low magnification scale bars, 500 μm. High magnification scale bars, 50 μm. Created in BioRender. Le Pichon, C. (2025) https://BioRender.com/g45x590.

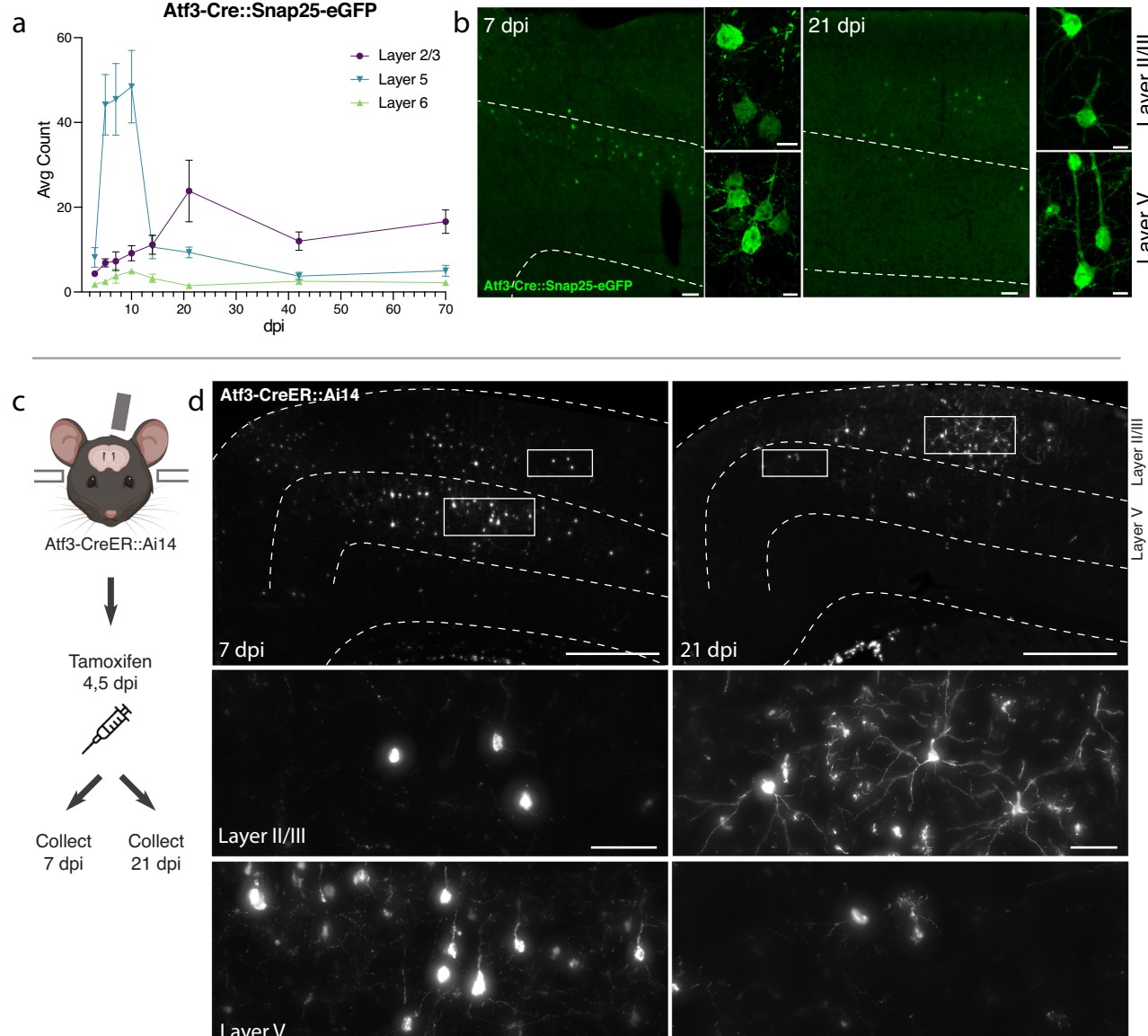

**Fig. 3 | Atf3-expressing neurons in layer V die, while those in layer II/III survive following mTBI. a** Quantification of the average count per section of Atf3-GFP neurons by cortical layer at 3, 5, 7, 10, 14, 21, 42, and 70 dpi. *N* = 6 per timepoint, 3–4 sections counted per animal. **b** Examples of Atf3-GFP endogenous labeling (left) at 7 and 21 dpi. Layer V is outlined. High magnification examples of GFP-immunolabeled neurons (right) in layer II/III (top) and layer V (bottom) at 7 and 21 dpi. Panels **a** and **b** use the Atf3-Cre::Snap25-eGFP mouse, which labels all neurons in which Atf3 has ever been expressed. **c** Schematic representation of tamoxifen dosing and tissue collection. The Atf3-CreER mouse was crossed to the

Ai14 RFP reporter (not neuron-specific). **d** Representative images of Ai14 signal in ipsilateral cortex at 7 dpi and 21 dpi. Layer V is outlined. Insets highlight neurons and other labeled cells in layer II/III and layer V. Insets for layer II/III suggest an earlier loss of projection complexity that is regained by 21 dpi. Inset for layer V at 21 dpi includes a neuron (left) and a glial cell (right). Panels **c** and **d** use the Atf3-CreER::Ai14 mouse, which labels any cell in which *Atf3* is activated in the presence of tamoxifen. For **a** error bars represent standard error of the mean (SEM). Scale bars: **b** Low magnification, 50 μm, high magnification, 10 μm; **d**. Low magnification, 500 μm, high magnification, 50 μm.

Next, we determined whether Atf3-GFP neurons displayed injury-induced alterations in excitability and basic membrane properties. We hypothesized the neurons would traverse a cellular state with altered electrophysiological properties reflecting the Atf3-response, and that this would differ between neurons from layer V versus layer II/III. We performed whole-cell patch clamp recordings of excitatory neurons from the neuron-specific Atf3-GFP mice at an acute (5–7 dpi) and late (21 dpi) timepoint. At the acute timepoint, layer II/III neurons expressed low levels of endogenous GFP and were thus too sparse and poorly defined to record from (Fig. 3b). Therefore, at this timepoint, we only recorded from GFP+ neurons in Layer V.

Layer V neurons at the acute stage were sufficiently healthy to record from, therefore not yet undergoing apoptosis, but they had clear alterations in electrophysiological properties compared to GFP-negative control neurons in the ipsilateral or contralateral cortex. They exhibited reduced intrinsic excitability: they were unable to sustain repetitive regular firing (Fig. 4a, b), had a higher rheobase (the minimal current required to initiate an action potential), and a significant decrease in the maximum number of spikes produced (Fig. 4c, d). They were also more depolarized (−54 mV vs. −68 mV in controls, Fig. 4e). However, their lowered excitability did not stem from alterations in passive membrane properties since these were no different from controls (input resistance, Supplementary Fig. 7f, and capacitance, which reflects cell size and/or arborization complexity, Fig. 4f).

GFP-negative neurons in the ipsilateral cortex were indistinguishable from contralateral uninjured neurons by nearly all measures, suggesting the decline in intrinsic excitability observed in GFP+ layer V neurons is a specific consequence in Atf3-positive neurons. Consistent with this, in our snRNAseq profiling of Atf3-captured neurons, we noted a pattern of downregulation of ion channels involved in action potential firing and maintenance of membrane potential (Fig. 4g). We confirmed the downregulation of Scn1a and Kcnq5 in GFP + layer V neurons in the tissue (Fig. 4h). The downregulation of these genes provides an explanation for the changes in excitability we observed in the Atf3-marked neurons.

At the later time point, 21 dpi, Atf3-GFP layer II/III neurons were able to maintain sustained firing, similar to GFP-negative controls (Fig. 4i, j, l). However, they exhibited increased tonic firing (Supplementary Fig. 7h), decreased rheobase threshold (Fig. 4k), increased input resistance (Supplementary Fig. 7j), and depolarized membrane potential (−53 mV vs. −66 mV in controls, Fig. 4m) - all measures that reflect increased excitability. GFP+ neurons also exhibited a reduced capacitance, suggesting that they might be more compact and/or less complex than GFP- controls (Fig. 4n). Interestingly, amplified hyperpolarization-activated (Ih) current, an inward current that is important in regulating action potential firing frequency, may contribute to enhanced tonic firing in GFP+ neurons in layer II/III[39,40] (Supplementary Fig. 7k, i). Together, these findings suggest that surviving layer II/III Atf3-GFP neurons adapt passive membrane properties to maintain sustained firing.

## Layer II/III neurons undergo axon initial segment reorganization following mTBI

Because we found that GFP+ neurons in layer II/III survive but are hyperexcitable, we wondered what the consequences would be on their output. We therefore examined whether their axon initial segment (AIS) underwent alterations. The AIS is a specialized structure at the base of the axon that is essential for generating action potentials[41,42]. Other neuron types transiently lose their AIS during regeneration[43–45]. Immunolabeling for Ankyrin-G, a master scaffolding protein of the AIS, suggested that the AIS in layer II/III neurons was lost at 7 dpi but regained by 14 dpi (Supplementary Fig. 8a–c). This transient loss of AIS markers was confirmed with staining for another AIS protein, β4-spectrin (Supplementary Fig. 8d). The hyperexcitability observed at 21 dpi in layer II/III neurons may be linked to a

reorganization of the AIS after mTBI. By contrast, layer V neurons did not lose their AIS (Supplementary Fig. 8a–d) − their reduced excitability may be due to lack of necessary machinery for ion flux caused by ion channel dysregulation (Fig. 4f). Previous studies of closed skull TBI have described an early loss of activity followed by a stage of hyperactivity[46]. Our observation of the transient disappearance of the AIS in layer II/III neurons may suggest that the observed changes in excitability are inherent to the recovery process. Interestingly, the AIS is also a site of neuronal polarization in development and regeneration[47].

## Atf3 is not required for mTBI-induced layer V neuron degeneration or death, but is required for downregulation of ion channels

Although Atf3 is required for regeneration of sensory neurons following peripheral nerve injury[12,48], our finding that Atf3-GFP cortical neurons in layer V die following mTBI suggested Atf3 may play a pro-degenerative role in the central nervous system. To investigate this, we deleted Atf3 in layer V neurons using an Rbp4-Cre driver (Rbp4-Cre::Atf3fl/fl, Atf3 cKO) and quantified degenerative pathology. We confirmed that layer V ATF3 expression was effectively reduced in Atf3 cKO mice at 7 dpi (Supplementary Fig. 9a, b), with any remaining ATF3+ nuclei likely representing non-neuronal cells. Layer V deletion of Atf3 did not affect dendrite degeneration[24] or presence of axonal swellings at 7 dpi, nor did it prevent mTBI-induced cell death or microgliosis (Supplementary Fig. 9d–i, 12c, d).

Because ATF3 drives downregulation of marker genes[11,12], we wondered if it is required for the loss of ion channel genes following mTBI (Fig. 4g, h). In situ hybridization of Scn1a in Atf3 cKO tissue (Rbp4-Cre::Atf3fl/fl::Sun1-GFP) revealed that the injury-induced loss of Scn1a in layer V neurons at 7 dpi is prevented by Atf3 cKO (Supplementary Fig. 10). This demonstration that downregulation of Scn1a is Atf3-dependent suggests that Atf3 could also drive transcriptional repression of additional ion channels leading to the observed functional deficits in Atf3-expressing neurons.

## The integrated stress response and SARM1 pathway are not required for mTBI-induced degeneration

Several interconnected signaling pathways influence neurodegeneration (Supplementary Fig. 11a). The integrated stress response (ISR), which halts translation in response to multiple neuronal stressors, and the SARM1 pathway, which controls Wallerian degeneration have both been shown to play neurotoxic roles in mTBI[6,49–53]. We wondered if we could protect layer V neurons by targeting these pathways. We use three genetic models to manipulate key players: a phospho-dead mutant of eIF2α (knock-in point mutation, serine to alanine substitution, eIF2a$^{S51A}$) that results in ~0% reduction of eIF2α and thus reduces ISR function[54], a conditional knockout of the pro-apoptotic effector of the ISR Ddit3 (Rbp4-Cre::Ddit3fl/fl, Ddit3 cKO) in layer V neurons[34], and a global knockout of the executor of Wallerian degeneration Sarm1[55] (Sarm1 KO). We find that targeting each of these pathways on its own is not sufficient to prevent dendrite degeneration, axon beading or swelling, or cell death (Supplementary Fig. 11). Therefore, we conclude that either none of these pathways is important for neuron death following mTBI, or that their collective action is required.

## Layer V deletion of Dlk prevents degeneration and death

We had previously noted the upregulation of multiple stress-responsive genes in deeper layer cortical neurons (Fig. 2g), so it was perhaps unsurprising that deletion of individual pathway effectors was not sufficient to prevent death of layer V neurons. We observed, however, that phosphorylated c-Jun (p-c-Jun), a known binding partner of ATF3, was distributed in a similar pattern as ATF3 in the ipsilateral cortex (Fig. 5a). These transcription factors, ATF3, and p-cJun, are known to be activated by the axon damage sensing protein DLK[56–58]

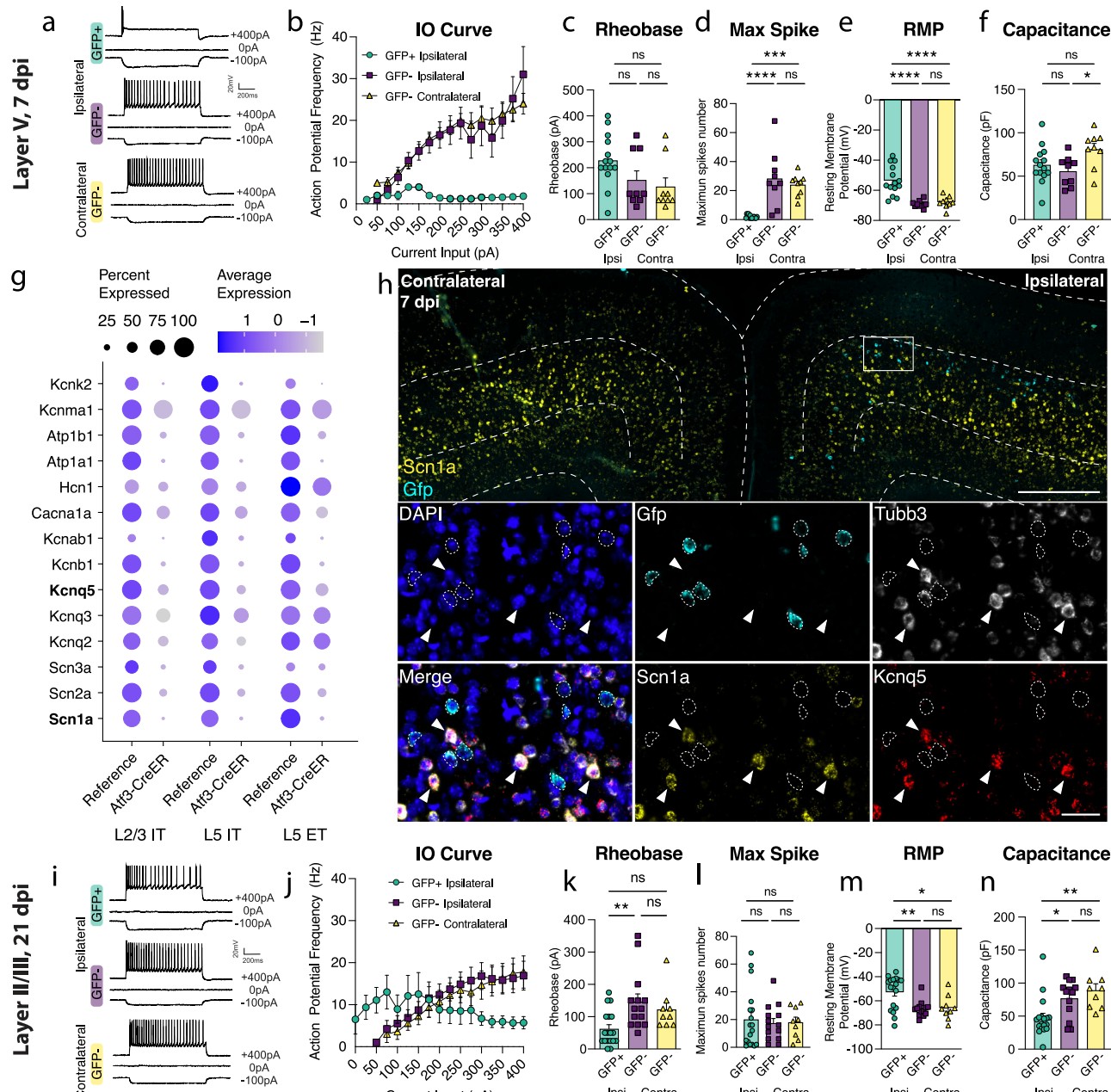

**Fig. 4 | Layer V Atf3-GFP neurons are unable to fire and downregulate ion channels while layer II/III Atf3-expressing neurons are electrophysiologically functional following mTBI. a** Examples of electrophysiological traces from 7 dpi layer V ipsilateral GFP+ and GFP- neurons and contralateral GFP- neurons. Quantifications of (**b**) IO curve, (**c**) rheobase, (**d**) max spike count (*** *p* = 0.0002, **** *p* < 0.0001), (**e**). resting membrane potential (**** *p* = <0.0001), and f. capacitance (* *p* = 0.0307) in 7 dpi layer V neurons. **g** Dot plot of select ion channels in the reference dataset compared to the Atf3-CreER dataset collected in this study showing dysregulation of ion channels in layer II/III and layer V neurons. Genes in bold were validated by in situ hybridization. **h** In situ hybridization validating downregulation of ion channels. Low magnification image of bilateral cortices from an Atf3-Cre::Snap25-GFP mouse at 7 dpi showing mRNA expression of Gfp (cyan) and Scn1a (yellow). Layer V is outlined. Inset shows mRNA of Gfp (cyan), Tubb3 (white), Scn1a (yellow), and Kcnq5 (red). Gfp+ neurons (outlined) lack expression of

Scn1a and Kcnq5 and have little to no expression of Tubb3. Arrowheads highlight Tubb3+ Gfp− neurons with high expression of Scn1a and Kcnq5. A single Z-plane is shown in the insets. **i** Examples of electrophysiological traces from 21 dpi layer II/III surviving ipsilateral GFP+ and GFP− neurons and contralateral GFP− neurons. Quantifications of (**j**) IO curve, (**k**) rheobase (** *p* = 0.0050), (**l**) max spike count, (**m**) resting membrane potential (* *p* = 0.0191, ** *p* = 0.0041), and (**n**) capacitance (* *p* = 0.0217, ** *p* = 0.0034) in 21 dpi layer II/III neurons. For **b** and **j** points represent the average of all neurons per group. Error bars represent SEM. For **c**–**f**, GFP+ ipsi *n* = 14 cells, GFP− ipsi *n* = 9, GFP- contra *n* = 9, each point represents one neuron recorded from *N* = 3 animals. For **k**–**n**, GFP+ ipsi *n* = 18 cells, GFP− ipsi *n* = 14, GFP-contra *n* = 9; each point represents one neuron recorded from *N* = 2 animals. ns: not significant. Significance was determined by Tukey's multiple comparisons test. Low magnification scale bar, 500 μm. High magnification scale bar, 50 μm.

(dual leucine zipper kinase; Supplementary Fig. 11a). DLK can drive ISR activity through phosphorylation of an ISR kinase, PERK[59,60] (Supplementary Fig. 11a, i–k), and promote Sarm1 activation through inhibition of its regulator, NMNAT[61–63] (Supplementary Fig. 11). We thus

reasoned that targeting DLK might be protective as a node sufficiently upstream of multiple neuronal stress responses. Furthermore, while the marker of DLK pathway activation p-c-Jun was detected in both layer V and layer II/III neurons, there was significantly higher

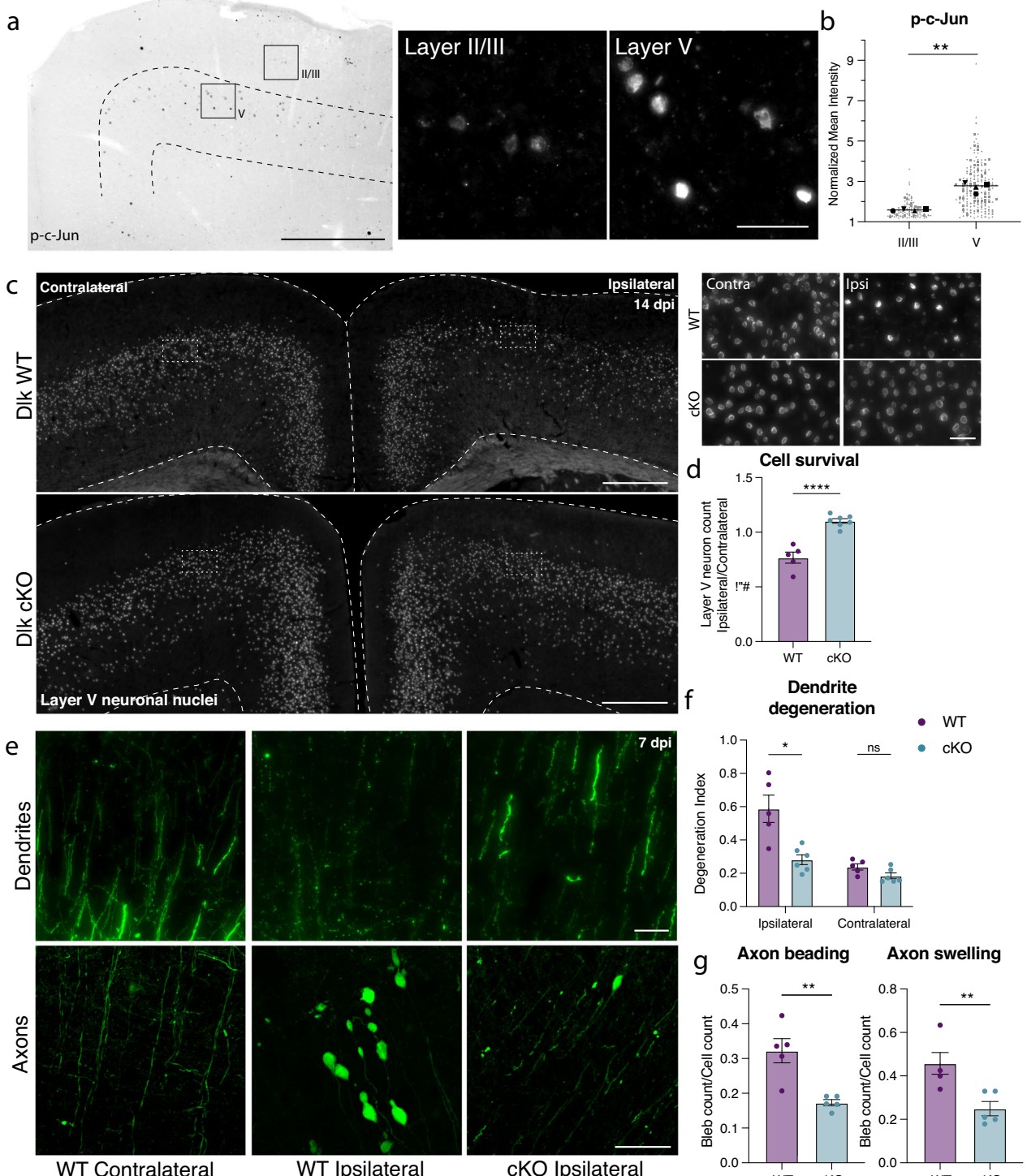

**Fig. 5 | Layer V neurons can be rescued from mTBI-induced death and degeneration by deletion of DLK. a** P-c-Jun immunolabeling in the ipsilateral cortex of WT mice at 7 dpi (left). Layer V is outlined. Insets show high magnification images of immunolabeling in layer II/III and layer V. **b** Quantification of p-c-Jun intensity in layer II/III and layer V at 7 dpi (**p = 0.0016 by two-tailed paired *t*-test). Only p-c-Jun+ cells are included based on a threshold of 1.2-fold expression compared to background. Average per animal and value per cell is displayed. Each shape represents one animal. N = 4, 2 sections per animal, 11–53 nuclei per animal, II/III nuclei n = 89, V nuclei n = 190. **c** Overview of ipsilateral and contralateral cortices in DLK WT and DLK cKO mice showing layer V GFP+ nuclei. Insets shown on the right. **d** Quantification of GFP+ neurons in ipsilateral compared to contralateral cortex in

WT (n = 5) and cKO (n = 7) mice at 14 dpi based on Sun1-GFP expression shown in **c** (** p = 0.0025 by two-tailed Mann–Whitney test). **e** High magnification images of YFP+ dendrites (top) and axons (bottom) in WT contralateral and ipsilateral cortex, and cKO ipsilateral cortex. **f** Quantification of dendrite degeneration at 7 dpi in WT (ipsi n = 5, contra n = 6) and cKO mice (ipsi n = 5, contra n = 6, **p = 0.0087 by two-tailed Mann–Whitney test). **g**. Quantification of axon beading (**p = 0.0079) and swelling (**p = 0.0079 by two-tailed Mann–Whitney test) at 7 dpi in WT (n = 5) and cKO (n = 5) mice. For **d**, **f**, **g**, each point represents the average of 3–4 sections per animal. Error bars represent SEM. Low magnification scale bars, 500 μm. High magnification scale bars, 50 μm.

expression in layer V neurons, correlating with their differential vulnerability (Fig. 5a, b). We thus tested if deleting DLK would promote survival of Atf3-neurons in layer V.

We conditionally deleted Dlk in layer V neurons using the Rbp4-Cre driver line (Rbp4-Cre::Dlk^fl/fl, Dlk cKO). We validated that *Dlk* transcript was selectively reduced in layer V DLK cKO neurons (Supplementary Fig. 12b, c). DLK deletion completely prevented layer V neuron death, rescuing the -15% loss of these neurons in the ipsilateral cortex (Fig. 5c, d). This rescue was maintained in Dlk cKO animals at 42 dpi (Supplementary Fig. 12d). By crossing the DLK cKO line with the Thy1-YFP reporter, we also observed that DLK cKO prevented mTBI-induced dendrite degeneration, and significantly reduced (but did not completely suppress) axon beading and swelling (Fig. 5e–g).

DLK signaling is essential in sensory neurons for recruitment of microglia and other inflammatory cells to sites of injury[64–66]. We investigated whether DLK cKO locally reduced microgliosis in the cortex after mTBI. We found that microgliosis was selectively reduced in layer V, where Dlk was depleted (Supplementary Fig. 13a, b). Interestingly, although DLK is required for *Csf1* upregulation in sensory neurons to recruit microglia following peripheral nerve injury, we found that CSF1 does not play a role in mTBI-induced cortical microgliosis (Supplementary Fig. 13e, f). Thus, mTBI-induced layer V microgliosis is not initiated through a neuronal injury response that actively recruits microglia via Csf1. Instead, microgliosis may occur as a response to factors released by apoptotic neurons, such as ATP, and prevention of neuron death by DLK deletion is therefore sufficient to prevent microglial recruitment.

We thus find that DLK activation is required for the degeneration of layer V neurons following mTBI, and that its differential activation in layer V and layer II/III neurons may be responsible for the differential vulnerability.

## Discussion

In this study, we develop and characterize a model of mild TBI in which we follow the cellular and molecular sequelae of a single impact to the skull on specific cortical neuron types during several weeks following injury. We used a closed-skull injury in order to model a clinically relevant mild trauma to the cortex. We find that despite its mild nature, this single-impact injury produces wide-ranging consequences to neurons within the cortex, from cell death to survival, with specific neuron types undergoing specific reproducible fates. We used the neuronal injury marker *Atf3* as a reporter to gain genetic access to a subset of neurons that transcriptionally responds to the injury. This allowed us to molecularly and spatially identify these neurons, describe their location and morphology, record their electrophysiological properties, and determine how their cellular states evolve over time. We discovered that the Atf3-responsive population of neurons falls into two categories: one located in layer V that undergoes neuronal death within 2 weeks after injury, and one in layer II/III that survives at least 2 months after injury. We found that the DLK signaling pathway is responsible for the death of the Atf3-responsive neurons in layer V, highlighting its value as a potential target for prevention treatments for TBI.

One challenge in studying models of mTBI is the ability to accurately quantify events or cell numbers using markers that are transient or altered by the injury itself. For example, commonly used markers to assess apoptosis are quite transient and may be missed. In this study, we demonstrate that many marker genes of cortical neuron types are lost after mTBI, making it impossible to accurately label or quantify the neuronal cell types in which they are normally expressed. This is consistent with other work showing that stress response mechanisms adopted by injured neurons to regain homeostasis after injury often result in the loss of expression of markers genes and proteins[11,12,67]. To overcome this, we relied on genetic labeling to track and record from our neurons of interest. This strategy also facilitated the enrichment of

this relatively rare population of cells within the cortex, allowing us to perform single nucleus transcriptomics to molecularly identify them using their transcriptome instead of individual marker genes. Additionally, this approach to study injured neurons is applicable to investigating the neuronal Atf3 response in a range of neurodegenerative conditions, beyond TBI.

The Atf3-reporter strategy highlighted the existence of two main populations of neurons after mTBI and a striking differential vulnerability between them. The neurons in layer V mostly died within 2 weeks of the injury while those in layer II/III survived. An intriguing aspect of the Atf3-GFP neurons in layer II/III is that they initially express the GFP reporter much more weakly than those in layer V. The initially faint reporter labeling may reflect a stronger translational repression in these neurons than in layer V which is reversed as they regain homeostasis. A question naturally arises: what is it about layer V neurons that makes them more vulnerable to degeneration? Layer V neurons have been highlighted as a vulnerable population within the cortex in multiple diseases and experimental models[4,68–72]. Using machine learning to classify the regeneration potential of neurons, a recent study identified layer V neurons as the least regenerative within the cortex[73]. It may be that their large somas, long axons, and metabolic demands require a molecular makeup that confers an increased vulnerability[74]. Selective vulnerability is seen across neurodegenerative diseases, necessitating further research to identify causative elements.

We found that DLK is essential for the death of layer V neurons, consistent with the known role of DLK signaling in promoting neuron death[56,57,75–80]. We speculate that the mild concussive TBI in our model produces axotomy of layer V projection neurons, and thus DLK-dependent death. While some layer II/III neurons have corticocortical projections, others project locally; thus, it is less clear whether they activate the Atf3-stress response pathway as a result of axotomy or another mechanism. Regardless, the majority of these layer II/III neurons do not degenerate within 70 days of the injury. Our snRNAseq data clearly demonstrate that despite sharing the expression of *Atf3* and *Ecel1*, the neurons from each layer express differential transcriptional programs, consistent with their divergent fates.

Recently, DLK inhibitors have been developed to treat neurodegenerative conditions[57,64,75,81–83], and understanding the role of the DLK pathway in mTBI will be critical for determining if it may be a viable therapeutic target. Recently, a Phase I clinical trial of a DLK inhibitor in ALS patients was halted after weeks of treatment due to observations of adverse effects including low platelet count, ocular toxicity, and altered touch sensation[84]. However, mTBI may be a more appropriate indication for trial as the injury timing can be known or even anticipated and dosing could be acute rather than chronic to reduce undesirable side-effects. Alternate methods to target DLK signaling may also be developed in future.

This study deepens our understanding of how cortical neurons respond to mTBI and reveals the heterogeneous nature of their responses, leading to differential vulnerability of distinct neuronal populations. It is possible that the initiation of neuronal stress responses after a single acute injury, as described in this study, can make surviving neurons susceptible to further injury, highlighting the need for targeted therapeutic approaches for populations prone to recurrent injury, such as athletes and military personnel. The identification of the DLK pathway as a potential target for neuroprotection opens new avenues for treatment strategies. Ultimately, these findings provide a foundation for future investigations into the heterogeneity of brain injury responses and potential clinical interventions aimed at mitigating the long-term impact of mTBI on cortical neurons.

### Limitations

One key limitation of this study is the reliance on a Cre-dependent labeling strategy, which requires functional translational machinery

and sufficient time for reporter expression. Using the Atf3-Cre line, we may have missed Atf3-positive neurons that are unable to produce the reporter and/or that die too rapidly. Our analyses using the inducible Atf3CreER mouse are limited to the specific times of tamoxifen delivery. The use of the Atf3-CreER line for snRNA sequencing of injured neurons also meant that we lacked the appropriate set of control nuclei, so we relied on a reference atlas to analyze the data. Finally, our conditional deletion experiments using Rbp4-Cre result in late embryonic deletion of genes which may have affected layer V neuron development. Of note, our focus on ATF3 induction in this study does not preclude the existence of additional injury programs in cortical neurons after mTBI.

## Methods

### Ethical compliance
Animal care and experimental procedures were performed in accordance with NICHD protocols 20-003 and 23-003 (Le Pichon lab) approved by the *Eunice Kennedy Shriver* National Institute of Child Health and Human Development ACUC.

### Mice
Mice were housed in rooms at 30–50% humidity with temperature around 72 °C and a 12 h light/dark cycle, with lights on from 6:00 a.m. to 5:59 p.m. Food and water were available *ad libidum*. Adult (>7 weeks of age) male and female mice were used for all experiments but were not analyzed separately. Thy1-YFP mice were acquired from The Jackson Laboratory (B6.Cg-Tg(Thy1-YFP)HJrs/J, Jax Stock No. 003782) Atf3-Cre mice, as previously described[11], were generated via knockin of an IRES-Cre sequence after the stop codon of Atf3 at the endogenous locus, such that endogenous Atf3 would remain intact. Atf3-CreER and Atf3 fl/fl mice were obtained from Dr. Clifford Woolf. Dlk fl/fl mice were obtained from Dr. Aaron DiAntonio. For sequencing studies, Atf3-CreER mice were crossed to Sun1-sfGFP (B6;129-Gt(ROSA)26Sortm5(-CAG-Sun1/sfGFP)Nat/J, Jax Stock No. 021039) and bred to heterozygosity for both alleles. For visualization of Atf3-expressing neurons, Atf3-Cre or Atf3-CreER mice were crossed to Snap25-LSL-eGFP (B6.Cg-Snap25tm1.1Hze/J, Jax Stock No. 021879) or Ai14 (B6.Cg-Gt(ROSA)26Sortm14(CAG-tdTomato)Hze/J, Jax Stock No. 007914) and bred to heterozygosity for both alleles. For all conditional knockout experiments, Rbp4-Cre mice (Rbp4-Cre (B6.FVB(Cg))-Tg(Rbp4-cre)KL100Gsat/Mmucd, MMRRC Stock No. 037128-UCD) were crossed to the flox line such that the floxed allele would be homozygous and Cre-negative littermates could be used as controls (this was for Dlk-fl, Atf3-fl and Csf1-fl lines). Chop fl mice (B6.Cg-Ddit3tm1.1Irt/J, Jax Stock No. 030816), Csf1 fl mice (shared by Dr. Sherry Werner[85]), and Sarm1 KO (C57BL/6J-Sarm1em1Agsa/J, Jax Sock No. 034399) were used. For Sarm1 KO, heterozygote littermates were used as controls, as heterozygous KO is not sufficient to prevent Wallerian degeneration. For ISR manipulation, we used Eif2-S51A mice (B6;129-Eif2s1tm1Rjk/J, Jax Stock No. 017601). Because homozygosity in this mutation is lethal, heterozygotes were used for experiments, and WT littermates were used as controls. For inducible Cre experiments, mice were dosed intraperitoneally with 75 mg/kg of a 20 mg/mL solution of tamoxifen mixed in corn oil at 4 and 5 dpi.

### Mild traumatic brain injury
Closed-skull mTBI was administered using the Leica Impact One (Leica Biosystems, Cat. No. 39463920) controlled cortical impact (CCI) device. Mice receiving injury were anesthetized with 2–2.5% isofluorane and positioned in a nose cone on foam pad. No stereotaxic restraint was used, however, neonatal ear bars were used to loosely stabilize the head to enhance consistency while maintaining movement upon impact. Mice were shaved and depilated around bregma. The 3 mm piston tip, mounted on the stereotax at an angle of 10° from the vertical plane, was centered roughly at bregma and moved 2 mm

lateral to the midline. The impactor was driven at a velocity of 5 m/s, depth of 1.5 mm, and dwell time of 200 ms. Animals receiving sham injuries were shaved, depilated, and anesthetized for the same amount of time as those receiving TBI, but were not administered the injury. Animals were given 5 mg/kg Meloxicam subcutaneously for analgesia immediately after injury and monitored after removal of anesthesia to evaluate righting reflex. Mice exhibiting tissue deformation following injury were excluded.

### Serum collection and neurofilament light simoa assay
Animals were lightly anesthetized using isofluorane until response to painful stimuli was lost. Blood was collected at baseline, 1, 9, and 14 dpi. The retro-orbital sinus of one eye was penetrated with a sterile unfiltered P1000 pipette tip. Blood was collected into BD Microtainer Capillary Blood Collector tubes (Cat. No. 365967) and allowed to clot at room temperature for 10 min. No more than 10% of the animal's body weight by volume was collected per session. Tubes were spun down at 6500 rpm for 10 min at 4 °C and the supernatant was aliquoted for storage at −80 °C. The Quanterix Neurology 4-plex A (Item 102153) assay was run following manufacturer instructions. Briefly, standards were plated in triplicate and test samples were plated in duplicate. Only neurofilament-light measurements were detectable and consistent between replicates. The average of the two replicates was reported as the final sample NfL measurement.

### Single nucleus RNA isolation and sequencing
Mice were anesthetized using 2.5% avertin and decapitated, and the brain was rapidly dissected. Ipsilateral cortical regions, roughly 4–5 mm diameter and centered around the injury site (as shown in Fig. 1a), were collected and rapidly frozen in pre-chilled tubes on dry ice, then stored at −80 °C. For the dissection of the desired cortical region, the ipsilateral hemibrain was rapidly isolated from the skull. Brain tissue anterior, posterior, and lateral to the area of interest (as visualized in Fig. 1b and Supplementary Fig. 4b) was removed, as well as any tissue below the corpus callosum. Nuclei isolation and sequencing were performed as previously described[86]. Ipsilateral cortical regions from 4 animals were pooled for each sequencing run to remove individual variability. Two datasets were integrated for the study, one collected from male animals and one from female animals. Sex differences were not observed. Samples were homogenized in a dounce homogenizer (Kimble Chase 2 ml Tissue Grinder) containing 1 ml freshly prepared ice-cold lysis buffer. The homogenate was filtered through a 40 μm cell strainer (FisherScientific #08-771-1), transferred to a DNA low bind microfuge tube (Eppendorf, #022431048), and centrifuged at 300 × *g* for 5 min at 4 °C. The washing step was repeated, and the nuclei were resuspended in 1× PBS with 1% BSA and 0.2 U/μl SUPERaseIn RNase Inhibitor (ThermoFisher, #AM2696) and loaded on top of a 1.8 M Sucrose Cushion Solution (Sigma, NUC-201). The sucrose gradient was centrifuged at 13,000 × *g* for 45 min at 4 °C for extra cleanup. The supernatant was discarded, the nuclei were resuspended in 1× PBS with 1% BSA, 0.2 U/μl SUPERaseIn RNase Inhibitor, and filtered through a 35 μm cell strainer (Falcon #352235). Before FACS sorting, 5 mM DRAQ5 (ThermoFisher #62251) was added to label nuclei.

GFP+DRAQ5+ nuclei were sorted and collected on a Sony SH800 Cell Sorter with a 100 mm sorting chip, and 10k GFP+ nuclei were loaded for sequencing. Using a Chromium Single Cell 3′ Library and Gel Bead Kit v3 (10X Genomics), GFP+ nuclei were immediately loaded onto a Chromium Single Cell Processor (10X Genomics) for barcoding of RNA from single nuclei. Sequencing libraries were constructed according to the manufacturer's instructions and resulting cDNA samples were run on an Agilent Bioanalyzer using the High Sensitivity DNA Chip as quality control and to determine cDNA concentrations. The samples were combined and run on an Illumina HiSeq2500. There were a total of 370 million reads passing the filter between the two

experiments (replicate 1 = 187,823,841, replicate 2 = 183,968,050). Reads were aligned and assigned to Ensembl GRm38 transcript definitions using the CellRanger v7.0.1 pipeline (10X Genomics). The transcript reference was prepared as a pre-mRNA reference as described in the Cell Ranger documentation.

## Single nucleus RNA sequencing data analysis
Following the CellRanger pipeline filtered sequencing data were analyzed using the R package Seurat version 4.1.3 following standard procedures. Outliers were identified based on the number of expressed genes (nFeature > 6000) and mitochondrial proportions (percent.mt > 5) and removed from the data. The data were normalized and scaled with the SCTransform function, dimensional reduction was performed on scaled data, significant principal components (PCs) were identified, and 30 significant PCs were used for downstream clustering. Clustering was performed using the Seurat functions FindNeighbors and FindClusters (resolution = 0.6). Clusters were then visualized with t-SNE or UMAP. Datasets were integrated with the IntegrateData function, and integrated data were then processed by the same methods. Data was visualized with the SCT assay or the RNA assay for dot plots, and plots were generated using Seurat functions. To assign cell types in an unbiased manner, sequenced nuclei were mapped onto a published and annotated mouse motor cortex snRNAseq reference dataset[32,87] using the Seurat MapQuery function. Clusters containing under 20 nuclei were removed from subsequent analyses. Comparisons to the nuclear reference dataset were made by merging it with our sequencing dataset and visualizing the RNA assay.

## Quantification of cell subtype similarity across the datasets
The similarity across cell subtypes between the reference and Atf3-CreER datasets was assessed using MetaNeighbor[88] v1.22.0 in R v4.3.3. After annotating the cells using reference mapping, cell subtypes and genes absent in one or both of datasets were removed. This pre-filtering process resulted in an input count matrix consisting of 17,642 genes and 7079 cells. MetaNeighbor is designed to quantify the degree to which cell subtypes replicate across the datasets based on the expression profiling of highly variable genes (HVGs). The HVGs were computed using the variableGenes function provided in Meta-Neighbor with default argument setting. Briefly, a gene was selected as an HVG if it was in the top quartile of variable genes and the top decile of expression bins for each dataset.

MetaNeighbor scored subtype-to-subtype similarity using the area under the receiver operator characteristic curve (AUROC). This computation was performed using the MetaNeighborUS function provided in MetaNeighbor with the fast_version and node_degree_normalization arguments set to TRUE. Technically, the function builds a cell network based on the Spearman correlation computed using the raw counts of HVGs between all pairs of cells in both datasets. Here, a node represents a cell, and an edge represents the strength of the correlations between nodes. For each cell (node), the cell subtype is predicted by accounting for the connectivity to neighboring cells using neighbor-voting algorithm. This algorithm creates a weighted matrix of predicted cell subtypes by performing matrix multiplication between the network and the binary vector (0, 1) indicating cell subtype membership. Afterward, this matrix is divided by the node degree, which returns a score for each cell equal to the fraction of its neighbors. The classification of cell subtypes is performed subsequently by computing the AUROC scores for each cell. This score is interpreted as the probability that the classifier correctly predicts that a cell subtype in membership ranked higher than that one not in membership. The AUROC ranges from 0 to 1, where 1 represents perfect classification and 0.5 represents a prediction as poor as random guessing. The output AUROC values are returned by averaging all pairs within a subtype.

## Fixed tissue harvest and immunostaining
Mice were anesthetized with 2.5% avertin and transcardially perfused with saline, followed by 4% paraformaldehyde. Tissue was post-fixed overnight and cryopreserved in 30% sucrose prior to sectioning. Thirty micrometer thick coronal slices were collected free-floating using a Leica CM3050 S Research Cryostat and stored in antigen preservation solution at 4 °C. For immunostaining, tissue was washed and permeabilized in 0.1% Triton-X100 in 1× PBS (PBSTx), then blocked in 5% normal donkey serum in 0.1% PBSTx. Primary antibodies were diluted in 0.5% normal donkey serum in 0.1% PBSTx and tissue was incubated overnight at 4 °C. Tissue was washed in 0.1% PBSTx and incubated in secondary antibody (ThermoFisher) diluted in 0.1% PBSTx for 1h, washed in 1× PBS, mounted on positively charged slides, and cover-slipped with Prolong Diamond (ThermoFisher #P36961). NeuroTrace (1:500, Life Tech. N21483) was applied following washes for 30 min. Primary antibodies: guinea pig anti-Ankyrin-G (1:500, Synaptic Systems, 386-005), rabbit anti-ATF3 (1:500, Novus Biologicals, NBP1-85816), rat anti-CD68 (1:500, Bio-Rad, MCA1957), rat anti-CTIP2 (1:500, abcam, ab18465), mouse anti-GFAP (1:500, Sigma-Aldrich, G3893), chicken anti-GFP (1:500, Invitrogen, A10262), chicken anti-IBA1 (1:500, Synaptic Systems, 234-006), rabbit anti-phospho-cJun Ser63 (1:300, Cell Signaling Tech., 9261), rabbit anti-phospho-H2AX (1:400, Cell Signaling Tech., 2577), rabbit anti-Olig2 (1:500, Millipore, AB9610). A custom-made rabbit anti-β4-Spectrin was shared by Dr. Damaris Lorenzo.

## Multiplexed in situ hybridization
Tissue was sectioned coronally at 16 μm onto positively charged slides using a Leica CM3050 S Research Cryostat. Slides were dried in the cryostat, then stored at −80 °C. Multiplexed in situ hybridization was performed according to the manufacturer's instructions for PFA fixed sections (ACD v2 kit). Probe targets were visualized using Opal dyes 520, 570, 690, or 780 (Akoya). Each in situ analysis was performed in at least $n$ = 2 mice. Probes: Atf3 (Cat. No. 426891), Atf4 (Cat. No. 405101), Ddit3 (Cat. No. 317661), Dlk (Cat. No. 458151), Ecel1 (Cat. No. 475331), Gad2 (Cat. No. 439371), Kcnq5 (Cat. No. 511131), Satb2 (Cat. No. 413261), Scn1a (Cat. No. 556181), Tubb3 (Cat. No. 423391).

## Electrophysiology brain slice preparation
Mice were anesthetized using Pentobarbital Sodium (NIH Veterinarian Services) and subsequently decapitated. Brains were swiftly removed and placed in an ice-cold cutting solution containing (in mM): 92 NMDG, 20 HEPES, 25 glucose, 30 NaHCO$_3$, 2.5 KCl, 1.2 NaPO$_4$ saturated, 10 Mg-sulfate, and 0.5 CaCl$_2$ with 95% O$_2$/5% CO$_2$. The solution had an osmolarity of 303–306 mOsm (Wescorp). The extracted brain was promptly blocked, dried on filter paper, and affixed to a platform immersed in ice-cold NMDG-based cutting solution within a chamber of a Leica VT1200 Vibratome. Coronal slices (300 μm thick) encompassing the somatosensory cortex, were cut at a speed of 0.07 mm/s. Post-slicing, sections were incubated in an NMDG-based cutting solution in a chamber for 5–10 min at 34 °C. Slices were then transferred to a chamber filled with a modified holding aCSF saturated with 95% O$_2$/5% CO$_2$. The solution contained (in mM): 92 NaCl, 20 HEPES, 25 glucose, 30 NaHCO$_3$, 2.5 KCl, 1.2 NaPO$_4$, 1 mM Mg-sulfate, and 2 mM CaCl$_2$, with an osmolarity of 303–306 mOsm, at room temperature for a minimum of 1 h. Slices were kept in the holding solution until being transferred to the recording chamber.

## Ex-vivo Whole-Cell Electrophysiology
Whole-cell patch-clamp electrophysiology studies were conducted following the methodology previously described[89]. Cells were visualized using infrared-differential interference contrast (IR-DIC) optics on an inverted Olympus BX5iWI microscope. The recording chamber was perfused at a flow rate of 1.5–2.0 ml per minute with artificial cerebrospinal fluid (aCSF) comprising (in mM): 126 NaCl, 2.5 KCl, 1.4

NaH2PO$_4$, 1.2 MgCl$_2$, 2.4 CaCl$_2$, 25 NaHCO$_3$, and 11 glucose (303–305 mOsm), using a pump from World Precision Instruments. For whole-cell recordings of intrinsic excitability, glass microelectrodes (3–5 MΩ) were employed, containing (in mM): 135 K-gluconate, 10 HEPES, 4 KCl, 4 Mg-ATP, and 0.3 Na-GTP. GFP-positive and GFP-negative cells were identified based on the presence or absence of GFP fluorescence in the DIC. Data were filtered at 10 kHz and digitized at 20 kHz using a 1440 A Digidata Digitizer (Molecular Devices). Series resistance (<20 MΩ) was monitored with a −5 mV voltage step. Cells exhibiting >20% change in series resistance were excluded from further analysis. For intrinsic excitability, following membrane rupture in voltage clamp, cells were transitioned to the current clamp configuration without holding current injection. Intrinsic excitability was evaluated by applying hyperpolarizing and depolarizing current steps (25 pA steps: 1 s duration), and changes in voltage and action potential firing were measured. Whole-cell recordings were conducted using a potassium gluconate-based internal solution. For all experiments, cells experiencing a 20% or higher increase in access resistance or high max spike rate consistent with interneurons were excluded from analysis.

### Imaging and quantifications
Images were collected using either a Zeiss slide scanner, Zeiss Axiocam 506, or Zeiss confocal LSM800. Images were quantified using FIJI. For cell counts, slide scanner images were cropped to equivalent contralateral and ipsilateral area for 3–4 sections and cells were counted by a blinded observer. For dendrite degeneration quantifications, 63X confocal images were collected from 3–4 sections (3 regions of interest (ROIs) per section per side). A FIJI macro was created to turn each image to binary and use the 'Analyze Particles' feature to collect circular area (circularity ≥ 0.2) and total area. For axon pathology area measurements, the magic wand tool in the Arivis Vision 4D software was used to manually select all axon blebs (YFP-high, DAPI-negative) for 3–4 sections per animal. Objects with a 3–10 μm$^2$ area were defined as axon beading, and those with area > 10 μm$^2$ were defined as axon swellings. For intensity quantifications, Z-planes were summed and ROIs were drawn around cells of interest based on either GFP expression, DAPI expression (specifically large nuclei > 40 μm$^2$ to select for neurons), or *Tubb3* expression for RNAscope, depending on the analysis. For normalized mean intensity, mean intensities were collected for background ROIs (negative for any signal), and cell mean intensity values were normalized to average background intensity. For percent area quantifications for IBA1 and GFAP, ROIs were drawn around the ipsilateral or contralateral cortex, an automated threshold (Triangle method) was set and images were turned to binary, then percent area was calculated using the 'Analyze Particles' feature. For all layer-specific quantifications, either DAPI or NeuroTrace was used to determine layers. Layer V was defined as the cortical layer with larger and more dispersed cells. Layer II/III was defined as everything above layer V, and layer VI is everything below.

### Statistics and reproducibility
Wherever possible, quantification of microscopy images was performed blinded. Statistical analyses were performed using GraphPad Prism 9, except in the case of mixed models analyses. No statistical method was used to predetermine sample size. Normality tests were performed and non-normal data were analyzed using non-parametric tests, as reported in figure legends. For supplementary figs. 10 and 11, in which our comparison of interest was *ipsi vs contra* in WT *vs cKO* (interaction), we used R package *nlme* and its function *lme* for the mixed effect modeling. The fixed effects were *Genotype*Side*, with a random effect being Side of each animal (i.e., *Side|animal*). We considered models with fixed variances among all the groups as well as models allowing for separate variances for either each Side and Genotype, or each Side and animal. Model selection criteria (AIC, BIC, and LRT) all pointed to models allowing for different variances as the best

(given that there are typically hundreds of values per each side of each animal). The comparison of interest was the interaction: change between cKO and wt genotypes in intensity differences between the sides, (ipsi−contra). For supplementary fig. 5, we were looking at differences between ipsi_pos & contra, and between ipsi_pos & ipsi_neg (GFP+ neurons and GFP- neurons of the ipsilateral and contralateral cortices; there is no contra±, just contra). To assess the differences between ipsi_pos & contra, and between ipsi_pos & ipsi_neg, we used R package *nlme* and its function *lme* for the mixed effect modeling. The fixed effects were the side and GFP status (i.e., ipsi_pos, ipsi_neg, and contra), with each one having a random component for each animal. Each data group (data for a given side and GFP status of each animal) was allowed to have its own variance. Reported are Holm-adjusted *p* values of Tukey contrasts for multiple comparisons of means.

### Reporting summary
Further information on research design is available in the Nature Portfolio Reporting Summary linked to this article.

### Data availability
The datasets generated during and/or analyzed during this study have been deposited in the Gene Expression Omnibus (GEO) under accession number GSE262317. Source data are provided with this paper.

### Code availability
Code used in analyzing single nucleus RNA sequencing data is available at https://github.com/malkasla/Alkaslasi-et-al._2024.

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

## Acknowledgements

We would like to thank Niamh Crowley for his help with the Quanterix assay, Dr. Vincent Schram and the NICHD Microscopy Core, Dr. Ryan Dale and the NICHD Molecular Genomics Core, Dr. Clifford Woolf, Dr. Aaron DiAntonio, and Dr. Sherry Werner for sharing mice, Dr. Damaris Lorenzo for sharing a custom-made antibody, and Drs. Alex Chesler, Mark Cookson, Mark Hoon, Ariel Levine, Tim Petros, and Nick Ryba for helpful discussions about the manuscript. Biorender.com was used to make figure graphics. CLP is intramurally funded by the National Institutes of Health, ZIA-HD008966 (NICHD). HAT is intramurally funded by ZIA MH002970-04 (NIMH), a NARSAD Young Investigator Award from the Brain and Behavior Research Foundation, and the BRAIN Initiative (RO1). HEY is funded by an NIH Center for Compulsive Behaviors Fellowship.

## Author contributions

M.R.A. and C.L.P. designed the experiments and wrote the manuscript. M.R.A. performed computational analyses. M.R.A., E.Y.H.L., A.S.G., and

H.S. performed data collection and image analysis. M.R.A., H.A.T., and H.E.Y. designed whole-cell patch clamp experiments. H.E.Y. and V.S.T. collected and analyzed electrophysiological recordings. M.S. and G.M. performed computational and statistical analyses.

## Funding

## Competing interests
The authors declare no competing interests.
