## [Transparent Peer Review file · Nature Communications]

The transcriptional response of cortical neurons to concussion reveals divergent fates after injury

Corresponding Author: Dr Claire Le Pichon

Version 0:

Reviewer comments:

Reviewer #1

(Remarks to the Author)

In this manuscript, “The transcriptional response of cortical neurons to concussion reveals divergent fates after injury” by Alkaslasi and colleagues, the authors explore how mild traumatic brain injury (mTBI) alters cortical transcription over the course of months following injury. The authors identify Activating Transcription Factor 3 (ATF3) as a gene upregulated in neurons after mTBI. ATF3-positive neurons also upregulate transcription of stress-related genes while downregulating expression of commonly used markers of neuronal sub-types. ATF3-positive neurons in L5 die after injury while neurons in L2/3 survive and continue to be able to drive APs. To investigate how ATF3-positive L5 neurons die, the authors showed that KO of MAP3K12 (DLK) attenuates L5 cell death. This study suggests targeting DLK after mTBI may be a novel therapeutic approach and demonstrates loss of a vulnerable cell-type after this subtle form of TBI. This study comes from a very well respected group known for their work on neurodegeneration and intracellular signaling, including DLK, in controlling cell death. Overall, this is an interesting and impactful study and uses novel genetic approaches to identify injury-disrupted cortical neurons and then combines both electrophysiology and snRNAseq to characterize the disruptions in these cells. The role of DLK in the pathology is quite exciting and potentially impactful. There are multiple issues, however, with the study design and presentation that need to be addressed before the study can fully be evaluated. Those are listed below, and if addressed, could lead to a very impactful study.

Major comments:

- The justification to study ATF3 in mTBI is not made clear in the introduction. A deeper explanation of ATF3, its role in neuronal injury, in particular in the CNS would be a welcome addition to this study.
- The authors make use of contralateral sections to as controls in many places in the study. In some places they use both contralateral sections and sham injured sections. While this has been a point of debate in the TBI field, in the last decade, the use of sham controls has generally been considered essential. While additionally using contralateral comparisons can be useful for assess injury/pathology focality, sham controls are the standard in the field. This is relevant to data on neuronal morphology in Fig 1. Emphasizing control to sham injury is the most relevant and rigorous comparison. Fig 1 J should quantify axon beading and axon swelling in shams. Fig 1i should make statistical comparisons to shams rather than/in addition to contra. The same is true in figure 4, Sup Fig 8 where all comparisons are made to contralateral or GFP- ipsilateral neurons.
- To say that cell loss of L5 cells is ‘progressive’ the authors need to show a statistical difference between 7 and 14 days post-injury, which they do not.
- In general, that statistical approaches used should be improved throughout. This includes 1) Only 2 animals were used to quantify Ddit3 in Sup Fig 5e and DLK in Sup Fig 11. A minimum of 3 animals should be used for any statistical analysis. 2) Were normality tests performed? Many of the data sets appear to not be normally distributed. Some electrophysiology data, IHC, and FISH data in particular should be tested for normality and then re-analyzed if necessary. 3) Average and SEM should be shown in Sup Fig 9 C for each data point. Also, because multiple measures are being made from the same animals, a post-hoc correction should be made. Alternatively, a mixed model approach could be used to statistically analyze this data. 4) What statistical approaches were used to quantify FISH when 10’s/100’s of cells were counted per animal? Do statistical measures hold up when n = animal average? Mixed models should also be considered here.
- Similarly, the details around snRNAseq are minimal in many places and should be better explained. This includes: 1) Tissue prep for snRNAseq is not clear. How was sample prepped? What cortical regions were used for snRNAseq? Was the contralateral tissue that was collected used for snRNAseq? The section on lines 135-140 should be greatly expanded. Even with what is included in the methods, many questions remain. 2) In figure 2F – what does down regulation “compared to the reference dataset” mean mathematically and statistically? What is the “reference dataset”? The same question applies

to Fig 4g. How was this reference data set normalized to the experimental data generated here? How was cluster identify confirmed between datasets? 3) How many technical and biological replicates of the snRNAseq experiment were performed? Was only one sample for WT and 1 sample for ATF3-GFP used? If so, that's really not sufficient. 3 technical replicates per condition is the standard in the field. 4). What quality control was performed on snRNAseq data? Methods mention outliers were identified based on the number of expressed genes and mitochondrial proportion of genes. More details are needed. 5) The description of snRNAseq data analysis needs to be greatly expanded. For example - What differential gene expression tools were used? How were the genes in Fig 2G selected?

- Does Atf3 actually cause downregulation of marker genes? Or is it just correlated with it? The fact that it does not affect loss of cells or other pathologies in L5 make this question especially relevant. Does conditional loss of ATF3 and/or DLK reduce downstream molecular changes (e.g. ion channels, inflammatory signaling, etc)?
- Why is are more cells labeled by ATF3-Cre in L2/3 3 days after injury? This doesn't make sense with a tamoxifen inducible Cre-based approach. If Cre is induced at 5 days post injury, all GFP labeled cells should already be labeled. In addition, more details on the promoters used to drive GFP expression would be helpful in interpreting increased GFP labeling of L2/3 cells over time. It is hard to understand why GFP levels would be so low in L2/3 in the first days after injury. But then rise so dramatically.
- The authors postulate that loss of AIS in L2/3 neurons 7 days post injury may underlie published studies showing decreased excitability. But they do not record from cells in L2/3 at this time, significantly weakening the argument.
- Published studies of ATF3 deficient mice show significant worse TBI outcomes in a weigh drop model (PMID: 29463176), as well as in ischemia (PMID: 35263513). How do the authors interpret such different findings from what they found with the CCI mTBI model?
- The loss of neurons after a single mTBI seems high compared to other studies. Can the authors comment on this? A very careful analysis of the cell loss data might be warranted, emphasizing the use of consistent methods throughout the study. Fig 1 uses Thy1-YFP cell counting. Fig 3 uses ATF3-Cre:SNAP-25-eGFP counting. Fig 5 uses contralateral/ipsilateral neuronal nuclei ratios. Sup Fig 9F uses total nuclei number. Sup Fig 10F uses ipsilateral/contralateral L5 neuron ratio. A consistent assay should be used throughout, and even better, multiple assays for each manipulation.
- There are multiple reporter mice used in different aspects of the study and can become quite confusing in terms of control experiments, etc. What is the ATF3-GFP mouse, how was it generated, and how does it work? This is not explained in the results section. Were all the lines confirmed to express GFP sufficiently at the time of tissue collections? This is particularly important for inducible expression of GFP in ATF3+ cells induced 5 days after injury and collected 7 days post-injury. Why is ATF3-Cre:SNAP25-eGFP used in Fig 3a/b but a different line is uses in c/d?

Minor comments:

- Number of neuronal nuclei sequenced does not match between line 143 and 153. Why is there a discrepancy?
- Method section on in situs is spaced differently than the rest of the doc.
- How was Iba1 % area calculated? Using a threshold method? If so, how was that calculated? Was the average Iba1 intensity per cell also decreased?
- In Sup Fig 6, why are IV curves shown as change in Vm on the y axis? This would be more rigorous, and standard, if it was simply shown as Vm.
- Is the increase in Ddit3 in ipsilateral GFP- versus contralateral GFP- cells scientifically meaningful? They appear identical.

Reviewer #2

(Remarks to the Author)

This study by Alkaslasi et al. characterizes a mild traumatic brain injury (mTBI) model and reports that this injury, which may be likened to a concussion, triggers differential injury responses across different cortical neuronal layers. Because ATF3 is clearly activated in response to mTBI in the injured hemisphere, the authors investigate this ATF3-inducing group of cells post-injury. They found that ATF3+ Layer V neurons die within 14 days post injury (dpi), while ATF3+ Layer II/III neurons persist and retain firing capabilities. DLK deletion rescues neuronal death in Layer V neurons, while genetic targeting of other individual stress response genes downstream of DLK does not.

This study employs an array of morphological, transcriptomic and electrophysiological tools to interrogate divergent neuronal responses to a mild traumatic brain injury, with the use of a list of genetic mutants to assess contribution of various stress response pathways particularly impressive. The findings provide new information about neuronal adaptations that correlate with surviving or dying neurons. In particular, ATF3 induction correlates with a population of layer V neurons that undergo degeneration and cell death after mTBI, although ATF3 itself is not required for this process, whereas DLK is required for such cell death. The apparent difference between layer II/III and layer V ATF3-inducing neurons in their injury responses is also interesting.

Main points:

Fig. 2f: Since the expression of neuronal subtype markers is downregulated after injury, how is the program mapping neurons to a reference atlas in order to accurately assign neuronal subtypes? A related comment on this point: even though previous studies showed downregulation of cell identity genes after peripheral nerve injury, a recent preprint on spinal cord injury reported no change in subtype marker genes among spinal (not brain) neurons following spinal cord injury (Skinnider et al., bioRxiv, 2023).

“The identification the DLK pathway as a potential target for neuroprotection opens new avenues for treatment strategies.” Applying genetic and pharmacological methods, a previous study (Welsbie et al., 2019) already reported a role for DLK and LZK in cell death signaling with a diffusive TBI model, although the cell type involved was retinal ganglion cells (RGCs).

Since the current study successfully recorded from mTBI-affected neurons, recording from the surviving DLK-cKO neurons, if possible, would lend increased novelty to the report that DLK mediates neuronal death.

Fig 4g: Although layer II/III neurons at 7dpi could not be recorded from, it would be useful to compare/contrast their ion channel gene regulation at this timepoint with Layer V neurons (i.e. add them to the dot plot).

Only ~10% (layer V) YFP-H positive neurons are ATF3 positive, indicating that ATF-inducing cells represents a small population of layer V neurons. Does this mean that only a small percent of layer V neurons respond to injury in this model? Could there be other injury responses that are not associated with ATF3 induction?

If ATF3 expression is very transient, this study could have missed other ATF3-inducing neurons. Please discuss the implications. In general, the conclusion on layer V is strong, but the conclusion on layer II/III feels weaker. Can the authors exclude any cell death from layer II/III? If so, could this be related to the mild nature of the injury such that more severe injury can still elicit cell death in layer II/III?

ATF3 cKO phenotype only scored in 7 dpi? What about later time point?

At first mention, please describe the detailed configuration of mutants in the Results section. For example: 1) eIF2 α (eIF2aS51A). Is this a knockin, point mutation? 2) Atf3-GFP: what is the configuration? Are there multiple configurations? There is some confusion: "For visualization of Atf3-expressing neurons, Atf3-Cre or Atf3-CreER mice were crossed to Snap25-LSL-eGFP ..." Figure 3, "Atf3-Cre::Snap25-eGFP" (not CreER?).

Multiple key experiments, such as DLK- and ATF3-conditional knockout (also Csf1? Not described in Methods), rely on the Rbp4-Cre driver that is activated during late embryonic development (~E16). How these deletions may affect developing Layer V neurons is not clear. Please discuss. If this represents a limitation to the study, please group this with other limitations in Discussion.

Axon beading and swelling did not seem to be completely suppressed in DLK cKO. Also, was cell death measured at multiple time points?

Results, Line 377-381: Is Csf1 the one and only means for "active microglial recruitment", such that the authors conclude, "layer V microgliosis is not initiated through a neuronal injury response that actively recruits microglia"? This experiment directly concludes more on Csf1 than the stated mechanism. Also, clarify if Csf1-cKO also relies on Rbp4-Cre.

Please clarify sequencing reads per sample.

Results, Lines 246-256: this paragraph feels a bit out of place here and seems generally redundant with Lines 183-203. Can these be streamlined or combined?

It would be helpful if more/most histology images (e.g., Fig. 1e-h) contained the "dpi" in the top right corner, as seen in Supplementary Figure 7a.

On figures:

Fig. 1b: It is not clear from the text, figure, or legend what timepoint this is.

Fig. 1d: Are the three shapes represent a different mouse each? A legend will be useful.

Fig. 1j: Lines 101-103: "We also observed beading of axons (fragments < 10 μ m²) representing axon degeneration. We found that both axon beading and axon swelling were increased only in the ipsilateral cortex at 7 dpi." One issue is that neither of these values reach statistical significance in the graph even though the difference appears large. Please clarify.

Fig. 3a: Quantification method: how are "upper" v "lower" Layer V neurons distinguished?

Fig. 3b: Text/legend state that GFP was immunolabeled, but it is not clear whether endogenous or immunolabeled GFP was used for quantifications in Fig 3a, or how the "lower expression of GFP" (Line 201-202) seen in 7-dpi neurons using immunolabeling may have affected those counts.

Fig. 3d: 21 dpi Layer V: without IHC markers, is it fair to confidently call these cells "a neuron" and "a microglion"? ("microglion" is a rarely used term)

Fig. 5d: Since "cell loss" is actually lower in the DLK cKO mouse, this graph title could be replaced with "cell survival".

Supp Fig. 5a: The graph x-axis should be adjusted to more accurately reflect the timepoints; if the first point is 3-dpi as it says in the legend, the point should land more to the right. Since most of the data in Supp Fig. 5 is from 7-dpi, the figure legend title could be changed to reflect this.

Supp Fig. S5f-g: How were GFP- cells identified as neurons for analysis?

Fig. S6a: Timepoint not clarified in any of the text.

Minor:

Line 213: Text references Supp. Fig. 5a, but likely means to reference Supp. Fig. 5b or c.

Line 215: Text references Supp. Fig. 5a,d but means to reference Supp. Fig. 5d. only?

Line 328: No panel S9j.

Reviewer #3

(Remarks to the Author)

Reviewer #4

(Remarks to the Author)

Around half of the population will experience an mTBI during their lifetime, however, the sequelae following these injuries at the cellular level have not been well characterized. Alkaslasi et al. use Atf3 as a transcriptional marker of injury responsive neurons in the motor cortex. Using a genetic labeling approach, they isolated Atf3 positive cells after mTBI and used scRNA-seq to map their identity to a reference atlas. Atf3+ cells included a subset of cortical neurons and glia. Atf3 expression was initially biased towards a subset of layer V neurons, which lost intrinsic excitability and degenerated. Atf3 expression was later observed in Layer II/III cortical neurons but these neurons generally did not degenerate in response to mTBI, suggestive of selective resilience of this population. Deletion of ATF3 activity did not prevent degeneration of cortical neurons after mTBI but deletion of Dlk did. Overall, these studies yield important insights into the cellular responses of cortical neurons to mTBI. Degeneration after a single mTBI event was layer and subclass specific, occurring primarily in a subset of Layer V cortical neurons.

Critiques:

-Figure 1i-k: The use of 3-4 biological replicates appears to be insufficient to determine significance as the trends are clear but this is not reflected in the quantification.

-Interpretation of scRNA-seq results are challenged by multiple factors:

- 1) Only genetically labeled Atf3 positive cells were profiled, thus there is a lack of internal controls. The assumption is made that Atf3 positive neurons represent the full set of injury responsive neurons, however, this may not be the case. It would be useful to collect a broader set of cells in the motor cortex after mTBI to validate the type-specificity of Atf3 expression and compare the transcriptional responses of Atf3 positive and negative cells.
- 2) The authors conclude that type-marker expression is downregulated in neurons post mTBI. This may be the case, however, the determination of type-specific gene expression is contingent on the accuracy of cell type mapping to the reference dataset, which can be challenging in injured neurons since their transcriptomic profiles have changed. To this end, in the discussion the authors state, "we demonstrate that many marker genes of cortical neuronal cell types are lost after mTBI, making it impossible to accurately label or quantify cell types in which they are normally expressed." Further evidence should be provided that mapping of transcriptomic data is robust, such as presenting confidence thresholds or presenting a larger set of molecular markers used to define the clusters. Even if the markers are degraded, type specific gene expression signatures should be observable, otherwise cells would be bioinformatically unmappable. In addition, usage of a second cell type assignment algorithm would be useful to demonstrate consistency.
- 3) Cell collections are underpowered to determine subtype-specificity. The Yao et al reference atlas for the motor cortex identified 116 cell types. The authors used broader subclass designations instead of full subtype identities, which is appropriate, but the term subtype is used throughout the study. Some groups are missing such as the L5 ET, however, it is unclear if this subclass was not mapped in this study because these cells are Atf3- or if they are not resolved due to sampling.
- 4) Since genetic labeling of ATF3 positive cells was done at 4-5dpi and the cells were collected at 7dpi, it is difficult to determine Atf3-dependent transcriptomic changes. This is illustrated by the lack of Atf3 expression in many collected cells, which is acknowledged by the authors. The authors should also acknowledge that the subset of Atf3+ cells at 7dpi may differ from the subset labeled at days 4-5.

-Figure 4: Different cell types naturally have different physiological properties, including types from within the same layer. How was it determined that Gfp+ and Gfp- recordings were done on cells of the same type?

-The lack of observed AIS disassembly in layer V neurons would seem to be incongruous with their reduced excitability and higher susceptibility. Do Atf3+ neurons in layer V maintain their AIS prior to degeneration? Discussion of this point could be further expanded upon.

-Deletion of Dlk prevented cell death after mTBI. It would be useful to demonstrate whether this treatment also prevented activation of injury responsive gene expression changes (e.g. Atf3, Ddit3, Atf4, etc.). Are protected cells physiologically normal?

Version 1:

Reviewer comments:

Reviewer #1

(Remarks to the Author)

Most of the concerns raised in the original review have been addressed. The remaining concerns that remain can be addressed with a small caveat section in the discussion. These include:

- consistent use of contralateral as control when sham injured animals are the standard in the field. This is especially important for mTBI models that can have diffuse injury in the contralateral cortex.
- Low n number (3 animals for multiple studies, 2 technical replicates for scRNAseq) for some experiments.

Reviewer #2

(Remarks to the Author)

Overall, the revision addresses many of the concerns expressed by the reviewers, including welcome updated and new figures. However, some issues remain to be clarified before publication.

1) Regarding the choice of focus on ATF3-positive cells (this was raised by the other reviewers too): some of the arguments and data in the rebuttal provides useful context and it would be helpful to incorporate these (perhaps as supp data) into the paper: e.g., Rebuttal page 25: "We were able to find Atf3-expressing cells, but they clustered separately and had no counterpart in the Sham data. This cluster, consisting of approximately 55 cells...".

2) The observation that ATF3 induction represents one of potentially a number of injury programs should be highlighted in both the Abstract and the main text, so that the significance is not inadvertently overinterpreted in the literature. Only about 10% of Layer V YFP neurons were found to be ATF3+ at 7 dpi, and at this time point, 15% of YFP neurons are already dead, but not necessarily because of ATF3. This point can be better addressed in Introduction-Paragraph 3 and certainly, in the Discussion - Limitations section. We also suggest that ATF3+ neurons always be identified as such, to avoid giving the impression that this injury-induced program may be applied more broadly than it can, e.g. "ATF3-Layer V neurons die within 14 days post injury (dpi), while ATF3-Layer II/III neurons persist and retain firing capabilities." Clarifying the percentage of ATF3 responsive neurons across layers / regions would be helpful.

3) In Rebuttal, in response to transient ATF3 expression after injury: "With our Cre-dependent genetic reporter strategy, even transient Atf3 expression should induce production of the reporter." This is not true for inducible Cre (CreER), where tamoxifen induction and ATF3 expression must coincide for CreER to be effective. Relevant to this: "We employed targeted snRNAseq of Atf3-expressing neurons using an inducible Atf3-IRES-CreER mouse line crossed to the INTACT nuclear envelope protein reporter" (Page 5). Also: "This mouse results from a cross between Atf3-IRES-Cre and a Cre dependent reporter line expressing GFP under control of the neuron-specific Snap25 promoter (Jax 021879)." Thus, both inducible and non-inducible forms of Cre were used in the study, and accordingly, whether Cre or CreER was used in a particular experiment should be stressed when presenting results. This is why the absence of a rigorous description of the ATF3-GFP strain in the original submission was so confusing, as both Cre and CreER mice were used in this study. Also, need to clarify if ATF3-IRES-Cre = ATF3-Cre and if ATF3-IRES-CreER = ATF3-CreER here. For CreER, the timing of Tamoxifen treatment is important when interpreting results.

4) Why does UMAP in new Fig. 2e looks so different from the old one? Annotation with a reference atlas should not change the shape of UMAP.

Minor:

Line 154: Missing % value for inhibitory neurons.

Reviewer #3

(Remarks to the Author)

Reviewer #4

(Remarks to the Author)

The authors have thoughtfully and carefully addressed my primary concerns and I do not have additional concerns. This manuscript reveals important neuronal populations specific effects after TBI.

Critique:

Figure S10i-k the image and quantification of Atf4 is missing.

REVIEWER COMMENTS

We thank the reviewers for their detailed assessments and helpful suggestions to improve the paper. We now provide a revised manuscript that includes new experimental data and figures, and new statistical analyses as requested. We respond below, point by point. We have uploaded a clean manuscript as well as a version highlighting changes to the original text.

New figures:

Supp fig 2f,
Supp fig 10
Supp fig 12d, e

Updated figures:

1i-k
2e-g
3a
4g
Supp 2a, c
Supp 3a
Supp 5a, e, g
Supp 9c
Supp 11d-e, i-j

Reviewer #1 (Remarks to the Author):

In this manuscript, “The transcriptional response of cortical neurons to concussion reveals divergent fates after injury” by Alkaslasi and colleagues, the authors explore how mild traumatic brain injury (mTBI) alters cortical transcription over the course of months following injury. The authors identify Activating Transcription Factor 3 (ATF3) as a gene upregulated in neurons after mTBI. ATF3-positive neurons also upregulate transcription of stress-related genes while downregulating expression of commonly used markers of neuronal sub-types. ATF3-positive neurons in L5 die after injury while neurons in L2/3 survive and continue to be able to drive APs. To investigate how ATF3-positive L5 neurons die, the authors showed that KO of MAP3K12 (DLK) attenuates L5 cell death. This study suggests targeting DLK after mTBI may be a novel therapeutic approach and demonstrates loss of a vulnerable cell-type after this subtle form of TBI. This study comes from a very well respected group known for their work on neurodegeneration and intracellular signaling, including DLK, in controlling cell death. Overall, this is an interesting and impactful study and uses novel genetic approaches to identify injury-disrupted cortical neurons and then combines both electrophysiology and snRNAseq to characterize the disruptions in these cells. The role of DLK in the

pathology is quite exciting and potentially impactful. There are multiple issues, however, with the study design and presentation that need to be addressed before the study can fully be evaluated. Those are listed below, and if addressed, could lead to a very impactful study.

Major comments:

- The justification to study ATF3 in mTBI is not made clear in the introduction. A deeper explanation of ATF3, its role in neuronal injury, in particular in the CNS would be a welcome addition to this study.

We have added to the introduction:

“Previous studies have observed neuronal Atf3 activation following injury to the central nervous system. Atf3 was activated in cortical neurons by TBI and in corticospinal neurons by axon transection depending on the proximity of the injury to the soma. Studies in Atf3-deficient mice found worse outcomes following TBI and ischemia, suggesting a protective role for Atf3, but did not distinguish between neuronal and glial activation of Atf3. We therefore hypothesized Atf3 would be activated in neurons after mTBI, but wondered how these neurons would compare to peripheral neurons in their ability to exhibit plasticity and recover. ” (lines 63-70)

- The authors make use of contralateral sections to as controls in many places in the study. In some places they use both contralateral sections and sham injured sections. While this has been a point of debate in the TBI field, in the last decade, the use of sham controls has generally been considered essential. While additionally using contralateral comparisons can be useful for assess injury/pathology focality, sham controls are the standard in the field. This is relevant to data on neuronal morphology in Fig 1. Emphasizing control to sham injury is the most relevant and rigorous comparison. Fig 1 J should quantify axon beading and axon swelling in shams. Fi 1i should make statistical comparisons to shams rather than/in addition to contra. The same is true in figure 4, Sup Fig 8 where all comparisons are made to contralateral or GFP- ipsilateral neurons.

We agree that the use of sham controls is important for the neuronal morphology data included in Figure 1. We have now added sham data for axon beading and swelling in Fig 1j. We have also added the statistical comparisons to sham in Fig 1 as suggested. Because neuron morphology was unchanged between sham and contralateral sections,

we are confident in our use of the contralateral tissue as an internal control for injury-induced changes in this model.

Due to the low-throughput nature of whole cell patch clamp and unavailability of our electrophysiologist collaborator within a reasonable timeframe, we did not include sham controls for the experiments in figure 4. However, we find that our contralateral neurons have electrophysiological measurements such as resting membrane potential and max spike number that are consistent with a previous study of uninjured animals (Scala et al., 2021, PMID: 33184512, Supplementary File 2), increasing our confidence that the contralateral sections serve as a reliable control.

- To say that cell loss of L5 cells is 'progressive' the authors need to show a statistical difference between 7 and 14 days post-injury, which they do not.

This has been corrected (we removed the word 'progressive').

- In general, that statistical approaches used should be improved throughout. This includes 1) Only 2 animals were used to quantify Ddit3 in Sup Fig 5e and DLK in Sup Fig 11. A minimum of 3 animals should be used for any statistical analysis.

A minimum of 3 animals is now used for all statistical analyses. We now show n=4 for Sup Fig 5e. We removed the statistical analysis for DLK WT vs cKO in situ since we did not readily have animals/tissue available to increase the n. However, we still include the data (Supp Fig 11X) because the n of 2 we had provided were consistent with DLK deletion, in addition to our many functional readouts of DLK cKO being protective of layer 5 neurons.

2) Were normality tests performed? Many of the data sets appear to not be normally distributed. Some electrophysiology data, IHC, and FISH data in particular should be tested for normality and then re-analyzed if necessary.

Normality tests have now been performed and non-parametric tests were used for non-normal data. This information was added to the methods (see the Statistical analysis section).

3) Average and SEM should be shown in Sup Fig 9 C for each data point. Also, because multiple measures are being made from the same animals, a post-hoc correction should be made. Alternatively, a mixed model approach could be used to statistically analyze this data.

We have corrected the graph in Sup Fig 9c to show average and SEM. As this analysis is a validation of the layer V knockout of ATF3, the only comparison of interest is within layer V, between WT and cKO, justifying the use of a t-test in this case.

4) What statistical approaches were used to quantify FISH when 10's/100's of cells were counted per animal? Do statistical measures hold up when n = animal average? Mixed models should also be considered here.

We now use mixed effect models and we thank the reviewer for that suggestion. (see figures and legends Supp Figs 5, 10, and 11, and Statistical analysis section in methods).

- Similarly, the details around snRNAseq are minimal in many places and should be better explained. This includes: 1) Tissue prep for snRNAseq is not clear. How was sample prepped? What cortical regions were used for snRNAseq? Was the contralateral tissue that was collected used for snRNAseq? The section on lines 135-140 should be greatly expanded. Even with what is included in the methods, many questions remain.

Sample prep included tamoxifen injection which is briefly described in lines 146-152 and in detail in lines 568-576. Contralateral tissue was collected but eventually disposed of, as it contained little to no GFP expression (Supp. Fig. 2b, since the Sun1-GFP marker is Cre-dependent) and thus was of no use for our snRNAseq approach that relied on Sun1-GFP for FACS to prior to snRNAseq. The tissue was rapidly collected and approximately subdivided based on the established injury site (Figure 1a,b). This is described in lines 569-574, that we have edited for additional detail and clarity:

“Ipsilateral cortical regions, roughly 4-5 mm diameter and centered around the injury site (as shown in Fig 1a), were collected and rapidly frozen in pre-chilled tubes on dry ice, then stored at -80 °C. For the dissection of the desired cortical region, the ipsilateral hemibrain was rapidly isolated from the skull. Brain tissue anterior, posterior, and lateral to the area of interest (as visualized in Figs 1b and Supp Fig 4b) was removed, as well as any tissue below the corpus callosum.”

The nuclei were then processed for FACS isolation of GFP+ nuclei, described in lines 586-591.

2) In figure 2F – what does down regulation “compared to the reference dataset” mean mathematically and statistically? What is the “reference dataset”? The same question applies to Fig 4g. How was this reference data set normalized to the experimental data generated here? How was cluster identify confirmed between datasets?

Reference mapping has recently become a standard method in the field of single cell RNA-seq to annotate cell types from new query datasets. This is a useful way to identify what cell types are present in a given dataset and is now possible for the mouse cortex thanks to large amounts of data that exist for this tissue and have been annotated (a consortium effort led by the Allen Brain Atlas).

The particular reference dataset we used is made up of single nuclei from 12 individual mice generated from the mouse primary motor cortex (a subset of the full data from ref 29 and ref 83; Azimuth HuBMAP Consortium). These underwent careful quality control and analyses to legitimize integrating them together to generate a reference dataset (ref 29). By comparing our data to this reference, we were able to query what type of cell each nucleus is from, based on its co-expression of transcripts that defines individual cell types in the reference. See also lines 614-618 and 621-647 in Methods.

Each dataset was internally normalized using the same parameters, and then merged with the other to show the data side by side (e.g. Fig 2f and Fig 4g). The data were not 'integrated' such that they are normalized to one another. However, when these data are displayed in a dotplot, the size and color of dots from each dataset are set to a common scale. The overall trend was that our data showed lower expression levels compared to the reference. This could have suggested that the reduction in gene expression in the Atf3CreER dataset was due to a difference in sequencing quality. However, we were reassured by the fact that not every gene in our dataset was lower expressed than in the reference dataset. For example, the blue-ness between datasets is similar for multiple genes in Fig 2f (e.g. Cux2, Gad2 in interneurons) and 4g (e.g. Hcn1 in L2/3, Kcnq3 in L5 ET).

Additionally, we were careful to validate claims made from these plots. For example, the lack of immunostaining for CTIP2 (Fig 2c; CTIP2 is the protein encoded by Bcl11b) is consistent with the reduction of Bcl11b by snRNAseq (Fig 2f), and the *in situ* hybridization of Scn1a and Kcnq5 (Fig 4h) confirms the reduction of these genes by snRNAseq (Fig 4g).

Note that Figures 2f and 4g have been updated to address other reviewer comments.

3) How many technical and biological replicates of the snRNAseq experiment were performed? Was only one sample for WT and 1 sample for ATF3-GFP used? If so, that's really not sufficient. 3 technical replicates per condition is the standard in the field.

WT animals were not included in this experiment, as the FACS-based method for isolation of Atf3-GFP nuclei requires GFP expression. 2 biological replicates were included, each consisting of nuclei pooled from 4 animals. (We considered that pooling 4 animals would average out individual variability.) There is no set standard in the single

cell transcriptomics field for number of replicates required in snRNAseq experiments. Considering the animal usage and the very high cost for such experiments, we look to the data to justify necessary replicates. Here, we were interested specifically in the type(s) of neuron that activate Atf3, and whether this was one specific population or whether it was made of multiple neuron types. Upon sequencing of 2 biological replicates, we found a strikingly similar representation of cell types (see below), which suggested we would not learn much more from further sequencing, at least considering the purpose to identify the cell types represented within the Atf3-expressing neurons.

4). What quality control was performed on snRNAseq data? Methods mention outliers were identified based on the number of expressed genes and mitochondrial proportion of genes. More details are needed.

Standard quality control filtering steps were performed on these data to exclude multiplets, as well as single nuclei that did not contain sufficient reads, or that showed excessive expression of reads mapping to the mitochondrial genome. Ambient mRNA from broken cells can often be detected as contaminants in single nucleus RNAseq experiments, so the standard in the field is to filter out any nucleus containing an excessive % mitochondrially encoded RNA. The specific numbers for filtering of feature expression and mitochondrial DNA have been added to the methods:

“Outliers were identified based on the number of expressed genes ($nFeature > 6000$) and mitochondrial proportions ($percent.mt > 5$) and removed from the data.”

5) The description of snRNAseq data analysis needs to be greatly expanded. For example - What differential gene expression tools were used? How were the genes in Fig 2G selected?

We selected genes known from the literature to be transcriptionally affected by injury. Differential gene expression analysis was not performed, as any differences in gene expression between clusters would likely be due to cell type identity, rather than transcriptional injury responses. As such, we elected to query a panel of previously known stress-responsive genes in Figure 2g, as described in the figure legend for Figure 2.

- Does Atf3 actually cause downregulation of marker genes? Or is it just correlated with it? The fact that it does not affect loss of cells or other pathologies in L5 make this question especially relevant. Does conditional loss of ATF3 and/or DLK reduce downstream molecular changes (e.g. ion channels, inflammatory signaling, etc)?

We agree with the reviewer that this is an important point. Gene repression could be caused by Atf3 or simply a correlate. Previous studies in sensory neurons have shown that Atf3 causes the downregulation of marker genes (see PMID: 32810432), so there is precedence for this hypothesis in another tissue. Additional studies outside of the nervous system have also shown that Atf3 can behave as an activator and as a repressor (PMID: 7515060, PMID: 9343434).

In new data using an Atf3 flox mouse line, we now show that *Scn1a* mRNA downregulation following mTBI is Atf3-dependent(see Supp Fig 10). This does not preclude the possibility that the downregulation of other genes is not Atf3-dependent. However, the example we provide demonstrates that downregulation of at least one gene can be explained by Atf3 activation, and suggests there are likely more.

- Why is are more cells labeled by ATF3-Cre in L2/3 3 days after injury? This doesn't make sense with a tamoxifen inducible Cre-based approach. If Cre is induced at 5 days post injury, all GFP labeled cells should already be labeled. In addition, more details on the promoters used to drive GFP expression would be helpful in interpreting increased GFP labeling of L2/3 cells over time. It is hard to understand why GFP levels would be so low in L2/3 in the first days after injury. But then rise so dramatically.

We assume the reviewer is referring to Figure 3a. In this plot we quantified the number of GFP expressing neurons over time in the Atf3-Cre::Snap25-eGFP mouse. (This is not the tamoxifen inducible CreER model that is shown in Fig 3c-d). In Fig 3a, any neuron that has expressed Atf3 should be labeled with GFP. We interpret the slow increase in layer II/III GFP-expressing neurons, from an average of ~5 per section at 3 dpi to an average of 20 at 21 dpi, as a gradual increase in the number of neurons activating Atf3

over time, with the neurons from the earlier time point surviving into the later time point (confirmed in Fig 3d). Because this is expression of a Cre-driven GFP dependent on the activation of an injury-induced transcription factor (Atf3), the expression of GFP must account for the time it takes for 1) Atf3 to be activated after injury, 2) Cre recombination to occur after Atf3 activation, 3) GFP translation to accumulate after recombination such that it can be readily detected. It is thus likely that the low expression at 3 days is due to insufficient time for GFP expression, or alternatively, that it is due to transient translational repression due to acute stress responses.

- The authors postulate that loss of AIS in L2/3 neurons 7 days post injury may underlie published studies showing decreased excitability. But they do not record from cells in L2/3 at this time, significantly weakening the argument.

We agree that recording from layer II/III neurons at 7 dpi would have added important insights, but it was not possible to confidently detect the endogenous fluorescence of these neurons on the electrophysiology rig to collect these recordings. We thus only suggest that the loss of AIS in these neurons may underlie the previously described decreased excitability, and hope that follow up studies can address this question.

- Published studies of ATF3 deficient mice show significant worse TBI outcomes in a weigh drop model (PMID: 29463176), as well as in ischemia (PMID: 35263513). How do the authors interpret such different findings from what they found with the CCI mTBI model?

Atf3 can be activated in macrophages, where it plays an anti-inflammatory role. Förstner et al. (PMID: 29463176) find that Atf3 regulates the expression of some chemokines and that, following TBI, Atf3 represses immune genes. In the absence of Atf3, they find increased presence of immune cells at the injury site, supporting an anti-inflammatory role for Atf3. Ma et al. (PMID: 35263513) find that Atf3 overexpression is similarly anti-inflammatory following middle cerebral artery occlusion, and they find decreased neuronal apoptosis in overexpression animals. While both of these studies compellingly indicate an important anti-inflammatory role for Atf3 that may be neuroprotective, neither study manipulates Atf3 in a cell type-specific way. It is possible that the effect in both studies, where Atf3 was germline deleted in all cell types, is due to the role of Atf3 in glia. In Ma et al., it may be that the anti-apoptotic effect of Atf3 overexpression is due to the reduction in inflammation driven by microglial Atf3. In our study, we delete Atf3 specifically in layer V excitatory neurons and find that it does not prevent cell death after mTBI. It is thus likely that the neuroprotective effect of Atf3 is not neuron-intrinsic. The role of Atf3 in microglia in this injury model would be very interesting to investigate in future studies.

- The loss of neurons after a single mTBI seems high compared to other studies. Can the authors comment on this? A very careful analysis of the cell loss data might be warranted, emphasizing the use of consistent methods throughout the study. Fig 1 uses Thy1-YFP cell counting. Fig 3 uses ATF3-Cre:SNAP-25-eGFP counting. Fig 5 uses contralateral/ipsilateral neuronal nuclei ratios. Sup Fig 9F uses total nuclei number. Sup Fig 10F uses ipsilateral/contralateral L5 neuron ratio. A consistent assay should be used throughout, and even better, multiple assays for each manipulation.

In a literature search for studies looking at neuron loss following mild TBI, we found varied results. While some studies report minimal to no cell death (Ogino et al., 2021), others report significant neuron death (Bu et al., 2016, Fig. 4; Gao et al., 2011, Fig. 2, 3; Raghupathi et al., 2002, Fig. 4; Sarkar et al., 2020, Fig. 6). It is challenging to conclude on the extent of cell death after mTBI across studies as it can vary greatly based on injury model, time point of analysis, location of analysis, and quantification method. While the loss of neurons in our study may differ from other studies, we found our quantification methods to be reliable and consistent.

In this study, we use one general approach to quantify neuron loss, and that is the quantification of neurons genetically labeled by a permanent reporter. We use multiple reporters to do this, including Thy1-YFP (sparse label of layer V neurons, figure 1), Atf3-Cre::Snap25-eGFP (neuron-specific label of cells that have activated Atf3, figure 3), and Rbp4-Cre::Sun1-sfGFP (nuclear label of all layer V excitatory neurons, figure 5). In the case of Atf3-Cre::Snap25-eGFP, we quantified the GFP expression over time in the ipsilateral cortex, as unilateral mTBI doesn't induce GFP expression in the contralateral cortex, and we found layer-specific loss of GFP-expressing neurons over time (figure 3a). In the case of Thy1-YFP and Rbp4-Cre::Sun1-sfGFP, we normalized the number of neurons in the ipsilateral cortex to the contralateral cortex, as we show in figure 1k that sham animals have roughly equivalent numbers of cortical neurons in the left and right hemispheres, and contralateral neurons do not exhibit degenerative pathology relative to sham (figure 1i, j). The quantification of ipsi vs. contra further provides an internal control to account for individual variation in cell counts. Both ipsi vs. contra approaches show approximately 25% loss of ipsilateral layer V neurons by 14 dpi (see figures 1k and 5d), increasing our confidence in the validity of the approach.

In sum, we carefully considered the issue of measuring cell loss, and we repeatedly found comparable results using orthogonal labeling methods, keeping our quantification method consistent throughout (counting the average number of positively labeled cells across 3-4 sections within the injured cortical region of each animal).

- There are multiple reporter mice used in different aspects of the study and can become quite confusing in terms of control experiments, etc. What is the ATF3-GFP

mouse, how was it generated, and how does it work? This is not explained in the results section. Were all the lines confirmed to express GFP sufficiently at the time of tissue collections? This is particularly important for inducible expression of GFP in ATF3+ cells induced 5 days after injury and collected 7 days post-injury. Why is ATF3-Cre:SNAP25-eGFP used in Fig 3a/b but a different line is used in c/d?

We agree our first description of this mouse in the results section was lacking a detailed explanation. We have now added the following text (line 195-199):

“To visualize and map the neurons that express *Atf3* after mTBI, we generated a neuron-specific *Atf3* reporter mouse (Atf3-GFP) in which GFP is permanently expressed only in neurons once *Atf3* is upregulated. This mouse results from a cross between Atf3-IRES-Cre and a Cre-dependent reporter line expressing eGFP under control of the neuron-specific Snap25 promoter (Jax 021879).”

In the study, we use 2 Cre lines and 3 reporters to visualize cells that express(ed) ATF3:

Cre driver lines: Atf3-IRES-Cre and Atf3-IRES-CreER (tamoxifen-inducible)

Cre-dependent reporter lines:

- LSL-Snap25-EGFP (cytoplasmic GFP restricted to neurons by the Snap25 promoter)
- Ai14 (LSL-TdTomato for cytoplasmic TdTomato under a universal promoter; expressed in all cells that turn on ATF3, not just neurons)
- LSL-Sun1-sfGFP (nuclear envelope-tethered GFP under a universal promoter, so expressed in all cells that turn on ATF3, not just neurons)

In Figure 3 it is true that we show results from experiments using 2 different crosses. Fig 3a-b uses the cross of Atf3-Cre x LSL-Snap25-eGFP to visualize and count all the *neurons* that express or expressed ATF3 at different time points after mTBI.

Fig 3c-d uses the cross of Atf3-CreER x Ai14 to visualize any cell (not just neurons) that was expressing ATF3 during the time of tamoxifen administration. Since we are specifically interested in the neurons here it would have been more ideal to use the Atf3-CreER line crossed to the LSL-Snap25-eGFP reporter line, but we did not have those mice available at the time. Nevertheless, the labeled neurons were straightforward to identify based on their size and morphology relative to the glial cells that were also labeled in the cross with Ai14. We hope this clarifies things for the reviewer.

Minor comments:

- Number of neuronal nuclei sequenced does not match between line 143 and 153. Why is there a discrepancy?

Thank you for pointing this out. This discrepancy has been corrected. It was due to a later adjustment to the reference mapping that we had forgotten to correct in both instances.

- Method section on in situs is spaced differently than the rest of the doc.

This has been corrected.

- How was Iba1 % area calculated? Using a threshold method? If so, how was that calculated? Was the average Iba1 intensity per cell also decreased?

Percent area for Iba1 was calculated as described in lines 731-734. We have added the thresholding method information to the methods:

“For percent area quantifications for IBA1 and GFAP, we used FIJI software. ROIs were drawn around the ipsilateral or contralateral cortex, an automated threshold (Triangle method) was set. Images were turned to binary, then percent area was calculated using the ‘Analyze Particles’ feature.”

Average Iba1 intensity per cell was not assessed in this study.

- In Sup Fig 6, why are IV curves shown as change in Vm on the y axis? This would be more rigorous, and standard, if it was simply shown as Vm.

“Change in Vm” was intended to reflect how Vm changes across cell categories. The values reported were Vm. The y-axis label has been corrected to prevent confusion.

- Is the increase in Ddit3 in ipsilateral GFP- versus contralateral GFP- cells scientifically meaningful? They appear identical.

In a new analysis with added n in Fig 5e, this is not significant.

Reviewer #2 (Remarks to the Author):

This study by Alkaslasi et al. characterizes a mild traumatic brain injury (mTBI) model and reports that this injury, which may be likened to a concussion, triggers differential injury responses across different cortical neuronal layers. Because ATF3 is clearly activated in response to mTBI in the injured hemisphere, the authors investigate this ATF3-inducing group of cells post-injury. They found that ATF3+ Layer V neurons die within 14 days post injury (dpi), while ATF3+ Layer II/III neurons persist and retain firing capabilities. DLK deletion rescues neuronal death in Layer V neurons, while genetic targeting of other individual stress response genes downstream of DLK does not.

This study employs an array of morphological, transcriptomic and electrophysiological tools to interrogate divergent neuronal responses to a mild traumatic brain injury, with the use of a list of genetic mutants to assess contribution of various stress response pathways particularly impressive. The findings provide new information about neuronal adaptations that correlate with surviving or dying neurons. In particular, ATF3 induction correlates with a population of layer V neurons that undergo degeneration and cell death after mTBI, although ATF3 itself is not required for this process, whereas DLK is required for such cell death. The apparent difference between layer II/III and layer V ATF3-inducing neurons in their injury responses is also interesting.

Main points:

Fig. 2f: Since the expression of neuronal subtype markers is downregulated after injury, how is the program mapping neurons to a reference atlas in order to accurately assign neuronal subtypes? A related comment on this point: even though previous studies showed downregulation of cell identity genes after peripheral nerve injury, a recent preprint on spinal cord injury reported no change in subtype marker genes among spinal (not brain) neurons following spinal cord injury (Skinnider et al., bioRxiv, 2023).

Although many key neuronal subtype markers are downregulated in *Atf3*-expressing neurons after injury, many are not. The several genes that are typically used as markers for the main neuron types in motor cortex are just a tiny fraction of their full transcriptomic identities. By using the full transcriptome of each subtype, we found that we are able to assign cellular identity to the single nuclei despite the downregulation of multiple markers after mTBI.

In our initial analyses, we looked at the top markers of each subtype in the reference dataset using the Azimuth RNA biomarkers tool and qualitatively assessed its specificity in the *Atf3*-CreER dataset. We found that many of the top 20 marker genes were still expressed in the *Atf3*-CreER dataset, albeit not those canonically considered markers

for a cell type. We include some examples here, showing screenshots of the top markers from Azimuth for the reference dataset next to feature plots of selected markers in the *Atf3CreER* data.

Sst

	auc	padj	pct_in	pct_out
Sst	0.949	1.88e-302	98	16.1
Lhx6	0.861	6.76e-271	81.6	9.9
Elfn1	0.902	6.84e-271	86.5	14
Rbp4	0.889	1.02e-250	84.5	14.3
9630002D21Rik	0.717	1.18e-247	45.7	2.67
Grin3a	0.958	2.25e-225	98.7	28.9
Reln	0.892	1.46e-222	86.5	17.7
Rab3b	0.932	2.35e-208	98.4	27
Npas1	0.78	4.48e-203	64.1	7.7
Kctd8	0.874	1.85e-196	86.5	18.7

L5 ET

	auc	padj	pct_in	pct_out
Pou3f1	0.971	0	97	14.1
Gm2164	0.948	0	92.6	8.7
Kcng1	0.945	0	93	9.04
Chst8	0.885	0	80.9	7.4
Adcyap1	0.86	0	75.3	4.92
Cacna1h	0.848	0	72.9	5.18
Fam84b	0.836	0	68.6	2.16
Npr3	0.821	0	66.2	2.59
Gm11730	0.723	0	44.8	0.186
Scn4b	0.891	1.64e-298	83.6	10.8

To legitimize our use of the Reference atlas, we now include a new analysis (Supp. Fig. 2f) to confirm the confidence of the cell type assignments, using a method called MetaNeighbor. MetaNeighbor (<https://doi.org/10.1038/s41467-018-03282-0>) measures how consistently cell types are replicated across datasets and allows for quick identification of clusters with high similarity. Using this package, we were able to quantify the similarity between clusters assigned using reference mapping of the *Atf3-CreER* dataset onto the reference dataset. We find a high degree of similarity for most cell types, with the assigned cell type having the highest confidence of accuracy.

Although Skinnider et al. report no change in subtype marker genes in spinal neurons after injury, there are many reasons why they might not observe Atf3-dependent downregulation. For one thing, they perform scRNAseq across all cell types in the spinal cord. Atf3-expressing cells, or those that downregulate markers, could represent a small population that they don't capture or miss in analysis. In fact, another paper that performed snRNAseq on spinal cord neurons following SCI, Matson et al. 2022, did report marker downregulation in an Atf3+ cluster (Figure 4, <https://doi.org/10.1038/s41467-022-33184-1>).

“The identification the DLK pathway as a potential target for neuroprotection opens new avenues for treatment strategies.” Applying genetic and pharmacological methods, a previous study (Welsbie et al., 2019) already reported a role for DLK and LZK in cell death signaling with a diffusive TBI model, although the cell type involved was retinal ganglion cells (RGCs). Since the current study successfully recorded from mTBI-affected neurons, recording from the surviving DLK-cKO neurons, if possible, would lend increased novelty to the report that DLK mediates neuronal death.

While we agree that recording from the surviving Dlk cKO neurons would be interesting, we are unable to perform these recordings in a way that would produce interpretable results. In our study, we record from ipsilateral Atf3-expressing neurons labeled by a GFP reporter and compare them to GFP-negative neurons (non-Atf3-expressing) in the ipsilateral and contralateral cortices. Because Atf3 is DLK dependent (Supp Fig 12e), crossing DLK flox mice to the Atf3-Cre line would not produce mice where Atf3-expressing neurons lack DLK.

An alternative to this approach would be to use our layer V Dlk cKO mice in which the Cre+ nuclei are labeled (figure 5c) and compare labeled ipsilateral Dlk cKO neurons to contralateral and/or sham neurons. The problem with this approach is that only a subset of layer V neurons activates Atf3 after injury and, because we know that ipsilateral Atf3-expressing neurons differ from other ipsilateral neurons in important ways, we would be unable to conclude whether the neurons recorded from would have been Atf3-expressing in the presence of Dlk. In fact, performing this experiment could give us the false impression that Dlk cKO neurons are similar to contralateral neurons just because we recorded from the wrong cells. Performing this experiment would require recording from 100s of neurons across multiple mice to be confident of the results. Still, we agree with the reviewer that recording from surviving Dlk cKO neurons after an injury would be of great interest, and might be better performed in an injury model in which a distinct and isolated population is consistently injured.

Fig 4g: Although layer II/III neurons at 7dpi could not be recorded from, it would be useful to compare/contrast their ion channel gene regulation at this timepoint with Layer V neurons (i.e. add them to the dot plot).

We agree and have added this data to the dotplot in figure 4g. We observe that ion channels are also downregulated in L2/3 IT, similar to Layer 5.

Only ~10% (layer V) YFP-H positive neurons are ATF3 positive, indicating that ATF-inducing cells represents a small population of layer V neurons. Does this mean that only a small percent of layer V neurons respond to injury in this model? Could there be other injury responses that are not associated with ATF3 induction?

We agree, there may be other injury responses not associated with ATF3 activated among layer V neurons following mTBI, but those are not addressed in this study. In a separate unpublished pilot study (female only, 1 replicate) in which we performed snRNAseq from neurons in the ipsilateral cortex at 7dpi using a Nestin-Cre driver line and Rbp4-Cre driver line (to enrich for layer V neurons), we have found that: 1) there were limited overall transcriptional changes between mTBI and sham (correlation plot below), 2) there were shifts in small populations of neurons that seem to reflect a mix of cell types but could be technical artifacts, 3) the only TBI-specific cluster was an *Atf3*-expressing cluster. This indicates that, if there are other injury responses activated, they are not nuclear and/or not transcriptional.

- 1) Correlation plot of gene expression (each dot = 1 gene) between TBI and Sham across each dataset. Trendline is in blue.

- 2) UMAP of Sham and TBI neurons in the ipsilateral cortex. Boxes highlight small clusters that shift slightly.

- 3) FeaturePlot of Atf3 - inset highlights Atf3+ cluster that appears in the TBI condition, but not Sham.

We ultimately chose not to include nor build upon this data in the manuscript because it would distract from the overall message. Our focus on the Atf3-expressing cells however, does not preclude the likely existence of additional injury responses that occur outside of the nucleus and/or at the translational or post-translational level.

If ATF3 expression is very transient, this study could have missed other ATF3-inducing neurons. Please discuss the implications. In general, the conclusion on layer V is strong, but the conclusion on layer II/III feels weaker. Can the authors exclude any cell

death from layer II/III? If so, could this be related to the mild nature of the injury such that more severe injury can still elicit cell death in layer II/III?

With our Cre-dependent genetic reporter strategy, even transient Atf3 expression should induce production of the reporter. We cannot exclude that there is any cell death in layer II/III, and we conclude only on the fate of Atf3-expressing neurons labeled by our reporter. It could be that a population of layer II/III neurons activates Atf3 acutely after injury and dies before reporter expression.

We now address this issue in a Limitations section at the end of our Discussion.

ATF3 cKO phenotype only scored in 7 dpi? What about later time point?

If we understand correctly, the reviewer is suggesting that we may have missed a partial effect of Atf3 cKO that may be evident at a later time point. We did perform morphology measurements on Atf3 cKO tissue at 14 dpi. The results indicated that Atf3 cKO remains ineffective even 2 weeks after impact, but the variability in our samples suggested a need for a higher sample size, and we did not have the tissue available to include this as conclusive data in the manuscript. We include it here for your review.

At first mention, please describe the detailed configuration of mutants in the Results section. For example: 1) eIF2 α (eIF2aS51A). Is this a knockin, point mutation? 2) Atf3-GFP: what is the configuration? Are there multiple configurations? There is some confusion: “For visualization of Atf3-expressing neurons, Atf3-Cre or Atf3-CreER mice were crossed to Snap25-LSL-eGFP ...” Figure 3, “Atf3-Cre::Snap25-eGFP” (not CreER?).

1) The eIF2a^{S51A} mouse (<https://www.jax.org/strain/017601>) (<https://www.jax.org/strain/017601>) was made by knocking in 3 base pair substitutions that result in a serine-to-alanine substitution (S51A).

2) See our response to Reviewer #1, included here:

“In the study, we use 2 Cre lines and 3 reporters to visualize cells that express(ed) ATF3:

Cre driver lines: Atf3-IRES-Cre and Atf3-IRES-CreER (tamoxifen-inducible)

Cre-dependent reporter lines:

- LSL-Snap25-EGFP (cytoplasmic GFP restricted to neurons by the Snap25 promoter)
- Ai14 (LSL-TdTomato for cytoplasmic TdTomato under a universal promoter; expressed in all cells that turn on ATF3, not just neurons)
- LSL-Sun1-sfGFP (nuclear envelope-tethered GFP under a universal promoter, so expressed in all cells that turn on ATF3, not just neurons)

In Figure 3 it is true that we show results from experiments using 2 different crosses. Fig 3a-b uses the cross of Atf3-Cre x LSL-Snap25-eGFP to visualize and count all the *neurons* that express or expressed ATF3 at different time points after mTBI.

Fig 3c-d uses the cross of Atf3-CreER x Ai14 to visualize any cell (not just neurons) that was expressing ATF3 during the time of tamoxifen administration. Since we are specifically interested in the neurons here it would have been more ideal to use the Atf3-CreER line crossed to the LSL-Snap25-eGFP reporter line, but we did not have those mice available at the time. Nevertheless, the labeled neurons were straightforward to identify based on their size and morphology relative to the glial cells that were also labeled in the cross with Ai14. We hope this clarifies things for the reviewer.”

Because 2 reviewers found the strains used in figure 3 to be confusing, we have added a line separating panels a and b from panels c and d and added more annotation of the mouse lines used, as well as included this clarifying information in the figure legend, in the hopes of making the figure more clear.

Multiple key experiments, such as DLK- and ATF3-conditional knockout (also Csf1? Not described in Methods), rely on the Rbp4-Cre driver that is activated during late embryonic development (~E16). How these deletions may affect developing Layer V

neurons is not clear. Please discuss. If this represents a limitation to the study, please group this with other limitations in Discussion.

For conditional knockouts, we only used the Rbp4-Cre driver line. The description of Csf1-flox being crossed to Rbp4-Cre has been added to the methods line 527-530.

The reviewer makes a good point (that we had also considered) that since the Rbp-Cre driver is embryonically (E16) the conditional deletions may have affected development of layer V. For example, DLK deletion can lead to a failure of developmental neuronal cell death of spinal motor neurons (PMID: 21893599). It is unknown whether this also occurs in the motor cortex, but it is possible. Other potential developmental alterations could occur in Rbp4-expressing neurons with deletion of DLK, Atf3, or Csf1. In the discussion we have now added mention of the caveat of using this driver line (see lines 497-498).

We had considered this caveat during our experimental planning, but despite its shortcomings, the Rbp4-Cre line was the best method we had available at the time to query DLK deletion in layer V (in the absence of reliable inducible CreER lines for layer 5 motor cortex). The advent of novel methods such as the definition of cell type specific enhancers will allow a more refined approach in future.

Axon beading and swelling did not seem to be completely suppressed in DLK cKO. Also, was cell death measured at multiple time points?

Beading and swelling was significantly reduced in DLK cKO but not completely suppressed (Fig 5g). We have added “but did not completely suppress’ in the result line 383. This is possibly because a part of the axonal pathology is a result of the primary injury (damage incurred by the initial impact), and not due to the DLK pathway that is initiated during the secondary injury response. It could also be that part of the pathology is due to additional signaling pathways.

We have performed additional experiments to quantify cell death in DLK cKO at the later time point of 42 dpi to confirm whether this is an enduring phenotype. This data is now included in Supplemental Figure 11d (and shown below). It shows that the cell

number still trends higher for the DLK cKO ($p=0.0571$ by t test for $n=3-4$).

Results, Line 377-381: Is *Csf1* the one and only means for “active microglial recruitment”, such that the authors conclude, “layer V microgliosis is not initiated through a neuronal injury response that actively recruits microglia”? This experiment directly concludes more on *Csf1* than the stated mechanism. Also, clarify if *Csf1*-cKO also relies on *Rbp4*-Cre.

We have rephrased this to reflect a conclusion only involving *Csf1*:

“layer V microgliosis is not initiated through a neuronal injury response that actively recruits microglia via *Csf1*”

Please clarify sequencing reads per sample.

This has been added to the methods section.

“There were a total of 370 million reads passing the filter between the two experiments (replicate 1 = 187,823,841, replicate 2 = 183,968,050).”

Results, Lines 246-256: this paragraph feels a bit out of place here and seems generally redundant with Lines 183-203. Can these be streamlined or combined?

Perhaps the reviewer was confused between the use of *Atf3*-Cre (original lines 183-203, now 195-205) and *Atf3*-CreER (original lines 246-256, now 264-274). We have tried to clarify it in the text and in our rebuttal (see line 300 of this rebuttal).

It would be helpful if more/most histology images (e.g., Fig. 1e-h) contained the “dpi” in the top right corner, as seen in Supplementary Figure 7a.

We agree and have added this for clarity.

On figures:

Fig. 1b: It is not clear from the text, figure, or legend what timepoint this is.

We have added time point information to the figure.

Fig. 1d: Are the three shapes represent a different mouse each? A legend will be useful.

This has been clarified in the figure legend.

Fig. 1j: Lines 101-103: “We also observed beading of axons (fragments < 10 μm^2) representing axon degeneration. We found that both axon beading and axon swelling were increased only in the ipsilateral cortex at 7 dpi.” One issue is that neither of these values reach statistical significance in the graph even though the difference appears large. Please clarify.

We have added n to this graph and the data are now statistically significant.

Fig. 3a: Quantification method: how are “upper” v “lower” Layer V neurons distinguished?

We have decided that this distinction is not relevant for the present manuscript, and have pooled them together as layer V.

Fig. 3b: Text/legend state that GFP was immunolabeled, but it is not clear whether endogenous or immunolabeled GFP was used for quantifications in Fig 3a, or how the “lower expression of GFP” (Line 201-202) seen in 7-dpi neurons using immunolabeling may have affected those counts.

Endogenous GFP was used to count GFP+ cells in Fig 3a, therefore these numbers may be underestimating the number that would be detected using anti-GFP (as shown in the examples in Fig 3b). Regardless, neurons in Figure 3a were all counted in the same manner and thus the layers and time points can be compared to each other. What matters here is the relative amounts, not absolute numbers as the main conclusion we take from Fig 3a is that within 1 week post-injury, many more Layer V neurons acutely express Atf3 than the other layers or at other time points.

Fig. 3d: 21 dpi Layer V: without IHC markers, is it fair to confidently call these cells “a neuron” and “a microglion”? (“microglion” is a rarely used term)

We are confident of the assignment of the neuron based on the neurite extensions clearly visible in the image. We have changed the figure legend text to refer to the putative microglion as a glial cell, as we agree with the reviewer that to unequivocally conclude this cell is a microglial cell would require labeling with a specific marker.

Fig. 5d: Since “cell loss” is actually lower in the DLK cKO mouse, this graph title could be replaced with “cell survival”.

We have made this edit to the figure.

Supp Fig. 5a: The graph x-axis should be adjusted to more accurately reflect the timepoints; if the first point is 3-dpi as it says in the legend, the point should land more to the right. Since most of the data in Supp Fig. 5 is from 7-dpi, the figure legend title could be changed to reflect this.

These edits have been made.

Supp Fig. S5f-g: How were GFP- cells identified as neurons for analysis?

GFP- cells were identified using DAPI and nuclear size greater than $45 \mu\text{m}^2$.

This information was added to the methods in lines 726-729:

“For intensity quantifications, Z-planes were summed and ROIs were drawn around cells of interest based on either GFP expression, DAPI expression (specifically large nuclei $> 40 \mu\text{m}^2$ to select for neurons), or *Tubb3* expression for RNAscope, depending on the analysis.”

Fig. S6a: Timepoint not clarified in any of the text.

We thank the reviewer for noticing this. Images in Supp Fig 6a are from 7 dpi.

Minor:

Line 213: Text references Supp. Fig. 5a, but likely means to reference Supp. Fig. 5b or c. This has been corrected to Supp Fig 5b, d

Line 215: Text references Supp. Fig. 5a,d but means to reference Supp. Fig. 5d. Only? This has been corrected to Supp Fig 5b, d.

Line 328: No panel S9j. Corrected to S9d-i

Reviewer #3 (Remarks to the Author):

Reviewer #4 (Remarks to the Author):

Around half of the population will experience an mTBI during their lifetime, however, the sequelae following these injuries at the cellular level have not been well characterized. Alkaslasi et al. use Atf3 as a transcriptional marker of injury responsive neurons in the motor cortex. Using a genetic labeling approach, they isolated Atf3 positive cells after mTBI and used scRNA-seq to map their identity to a reference atlas. Atf3+ cells included a subset of cortical neurons and glia. Atf3 expression was initially biased towards a subset of layer V neurons, which lost intrinsic excitability and degenerated. Atf3 expression was later observed in Layer II/III cortical neurons but these neurons generally did not degenerate in response to mTBI, suggestive of selective resilience of this population. Deletion of ATF3 activity did not prevent degeneration of cortical neurons after mTBI but deletion of Dlk did. Overall, these studies yield important insights into the cellular responses of cortical neurons to mTBI. Degeneration after a single mTBI event was layer and subclass specific, occurring primarily in a subset of Layer V cortical neurons.

Critiques:

-Figure 1i-k: The use of 3-4 biological replicates appears to be insufficient to determine significance as the trends are clear but this is not reflected in the quantification.

We have added new data to increase the sample size to 5-7 for these analyses. Measurements that were previously trending are now significant.

-Interpretation of scRNA-seq results are challenged by multiple factors:

1) Only genetically labeled Atf3 positive cells were profiled, thus there is a lack of internal controls. The assumption is made that Atf3 positive neurons represent the full set of injury responsive neurons, however, this may not be the case. It would be useful to collect a broader set of cells in the motor cortex after mTBI to validate the type-specificity of Atf3 expression and compare the transcriptional responses of Atf3 positive and negative cells.

We agree with the reviewer that collecting a broader set of cells in the motor cortex in mTBI and sham animals would allow for investigation of the transcriptional programs across cell types, and had started our study with this goal. We initially performed snRNAseq using the pan-neuronal Nestin-Cre driver line (female only, 1 replicate), but found no major transcriptional differences between TBI and Sham animals. Knowing that we should at least find Atf3 expression in this dataset, we thought affected neurons might represent too small a population among all cortical neurons to isolate with snRNAseq. We then decided to enrich for layer V neurons by isolating neurons with the

Rbp4-Cre driver. This approach resulted in the data shared above in response to Reviewer #2, that we share again here.

We were able to find Atf3-expressing cells, but they clustered separately and had no counterpart in the Sham data. This cluster, consisting of approximately 55 cells, appeared to represent a mixed population of different cell types. Two previous studies (PMID: 31592768, PMID: 32810432) showed that Atf3 activation in sensory neurons leads to such a profound change in gene expression that Atf3-expressing cells cluster with each other rather than with their respective cell types, which we interpreted to be occurring in our data as well. So although we could identify Atf3-expressing cells, we could not directly compare them to their Atf3-negative counterparts.

Looking beyond the Atf3+ cluster, there were shifts in small populations of neurons that seem to reflect a mix of cell types or could be technical artifacts.

When we performed differential expression analysis between broader groups of cells in TBI and sham, we found little to no differentially expressed genes.

We ultimately chose not to include nor build upon this data in the manuscript because it would distract from the overall message. Our focus on the Atf3-expressing cells however, does not preclude the likely existence of additional injury responses that occur outside of the nucleus and/or at the translational or post-translational level.

2) The authors conclude that type-marker expression is downregulated in neurons post mTBI. This may be the case, however, the determination of type-specific gene expression is contingent on the accuracy of cell type mapping to the reference dataset, which can be challenging in injured neurons since their transcriptomic profiles have changed. To this end, in the discussion the authors state, “we demonstrate that many marker genes of cortical neuronal cell types are lost after mTBI, making it impossible to accurately label or quantify cell types in which they are normally expressed.” Further evidence should be provided that mapping of transcriptomic data is robust, such as presenting confidence thresholds or presenting a larger set of molecular markers used to define the clusters. Even if the markers are degraded, type specific gene expression signatures should be observable, otherwise cells would be bioinformatically unmappable. In addition, usage of a second cell type assignment algorithm would be useful to demonstrate consistency.

Same response as to reviewer 2 above:

Although many key neuronal subtype markers are downregulated in Atf3-expressing neurons after injury, many are not. The several genes that are typically used as markers for the main neuron types in motor cortex are just a tiny fraction of their full transcriptomic identities. By using the full transcriptome of each subtype, we found that we are able to assign cellular identity to the single nuclei despite the downregulation of multiple markers after mTBI.

In our initial analyses, we looked at the top markers of each subtype in the reference dataset using the Azimuth RNA biomarkers tool and qualitatively assessed its specificity in the Atf3-CreER dataset. We found that many of the top 20 marker genes were still expressed in the Atf3-CreER dataset, albeit not those canonically considered markers for a cell type. We include some examples here, showing screenshots of the top markers from Azimuth for the reference dataset next to feature plots of selected markers in the Atf3CreER data.

Sst

	auc	padj	pct_in	pct_out
Sst	0.949	1.88e-302	98	16.1
Lhx6	0.861	6.76e-271	81.6	9.9
Elfn1	0.902	6.84e-271	86.5	14
Rbp4	0.889	1.02e-250	84.5	14.3
9630002D21Rik	0.717	1.18e-247	45.7	2.67
Grin3a	0.958	2.25e-225	98.7	28.9
Reln	0.892	1.46e-222	86.5	17.7
Rab3b	0.932	2.35e-208	98.4	27
Npas1	0.78	4.48e-203	64.1	7.7
Kctd8	0.874	1.85e-196	86.5	18.7

L5 ET

	auc	padj	pct_in	pct_out
Pou3f1	0.971	0	97	14.1
Gm2164	0.948	0	92.6	8.7
Kcng1	0.945	0	93	9.04
Chst8	0.885	0	80.9	7.4
Adcyap1	0.86	0	75.3	4.92
Cacna1h	0.848	0	72.9	5.18
Fam84b	0.836	0	68.6	2.16
Npr3	0.821	0	66.2	2.59
Gm11730	0.723	0	44.8	0.186
Scn4b	0.891	1.64e-298	83.6	10.8

To legitimize our use of the Reference atlas, we now include a new analysis (Supp. Fig. 2f) to confirm the confidence of the cell type assignments, using a method called MetaNeighbor. MetaNeighbor (PMID: 29491377) measures how consistently cell types are replicated across datasets and allows for quick identification of clusters with high similarity. Using this package, we were able to quantify the similarity between clusters assigned using reference mapping of the Atf3-CreER dataset onto the reference dataset. We find a high degree of similarity for most cell types, with the assigned cell type having the highest confidence of accuracy.

3) Cell collections are underpowered to determine subtype-specificity. The Yao et al reference atlas for the motor cortex identified 116 cell types. The authors used broader subclass designations instead of full subtype identities, which is appropriate, but the term subtype is used throughout the study. Some groups are missing such as the L5 ET, however, it is unclear if this subclass was not mapped in this study because these cells are Atf3- or if they are not resolved due to sampling.

We used the term 'subtype' as a general term to indicate a type within a type, but agree this is confusing with the nomenclature used in studies that annotate large datasets. We have now replaced it with the term 'subclass'.

It is always possible that further sequencing could identify additional subclasses that activate Atf3 that are maybe more rare and were not captured in the ~8,000 cells we sequenced. Our goal was to determine whether multiple neuronal subclasses activated Atf3, or whether this population was dominated by a few subclasses. Although we cannot know whether the data we collected represent the full diversity of the Atf3-expressing population, we are confident that the subclasses we identify do activate Atf3.

The reference dataset in our original analyses was acquired prior to the full publication of Yao et al (it was a preprint). For some reason, L5 ET was not in that reference dataset. In new analyses to address other reviewer comments, we use an updated version of this dataset. L5 ET is now represented in our Atf3CreER dataset.

Note that Figure 2e shows all subclasses mapped in our data, but that L5/6 NP, L6b, and Sncg were removed from further analyses because they were represented by fewer than 10 nuclei.

4) Since genetic labeling of ATF3 positive cells was done at 4-5dpi and the cells were collected at 7dpi, it is difficult to determine Atf3-dependent transcriptomic changes. This is illustrated by the lack of Atf3 expression in many collected cells, which is acknowledged by the authors. The authors should also acknowledge that the subset of Atf3+ cells at 7dpi may differ from the subset labeled at days 4-5.

We do not intend to show Atf3-dependency in our snRNAseq experiments, we only use Atf3 as a reporter to isolate neurons undergoing a transcriptional injury response. While Atf3 has been described as a master regulator of this injury response in other contexts, it is likely that many transcriptional changes here are not Atf3-dependent.

We fully acknowledge that the subset of Atf3+ cells labeled at 4-5 using the Atf3-CreER strategy and visualized at 7dpi may differ from the total population of Atf3+ neurons at 7

dpi. We did not intend to suggest otherwise. We have rephrased a sentence that may have caused this confusion. (“At 7 dpi, most labeled neurons were located in layer V, while those remaining at 21 dpi were primarily found in layer II/III.” Line 270)

-Figure 4: Different cell types naturally have different physiological properties, including types from within the same layer. How was it determined that Gfp+ and Gfp- recordings were done on cells of the same type?

It is impossible to definitively determine that we were recording from the same cell types. Based on our snRNAseq data, we know that the GFP+ neurons represent a wide variety of neuron types, and thus we decided that random comparison to GFP- neurons was the best we could do for the control.

Scala et al., 2020 (PMID: 33184512) showed that while broad transcriptomic neuronal types (Sst, Pvalb, etc) were physiologically distinct from one another, subclasses within these families were more similar to one another. They show that excitatory neuron subtypes within and between layers share many electrophysiological properties (ED Fig. 4, Supp. File 2). Comparing our data to theirs, and validating our approach, we find that our recordings (from Gfp+ or - cells) are most consistent with excitatory neurons from either layer V or layer II/III based on rheobase, max spike, input resistance, and resting membrane potential.

-The lack of observed AIS disassembly in layer V neurons would seem to be incongruous with their reduced excitability and higher susceptibility. Do Atf3+ neurons in layer V maintain their AIS prior to degeneration? Discussion of this point could be further expanded upon.

Atf3+ neurons in layer V do maintain their AIS prior to degeneration, as shown in Supp Fig 8a-b. While we postulate for layer II/III that transient loss of activity that has been previously reported in TBI may be linked to the transient loss of the AIS, this is not the only way in which neuronal activity can be affected. Layer V Atf3-expressing neurons displayed dysregulation of many ion channel genes, suggesting that their reduced excitability may be due to lack of the necessary machinery for ion flux.

This has been addressed in the manuscript, with the following addition to lines 329-331:

“By contrast, layer V neurons did not lose their AIS (Supplementary Fig. 8a-d) — their reduced excitability may be due to lack of necessary machinery for ion flux caused by ion channel dysregulation (Figure 4f).”

-Deletion of Dlk prevented cell death after mTBI. It would be useful to demonstrate

whether this treatment also prevented activation of injury responsive gene expression changes (e.g. Atf3, Ddit3, Atf4, etc.). Are protected cells physiologically normal?

Dlk cKO did prevent activation of injury responsive gene expression changes. We have performed new experiments and added new data showing in Supp. Fig. 10i-k that Ddit3 and Atf4 are reduced by Dlk cKO, and in new data added to Supp. Fig. 12e, we show that ATF3 is reduced by Dlk cKO. We cannot conclude on whether protected cells are physiologically normal, as described in response to Reviewer #2, included again below.

“While we agree that recording from the surviving Dlk cKO neurons would be interesting, we are unable to perform these recordings in a way that would produce interpretable results. In our study, we record from ipsilateral Atf3-expressing neurons labeled by a GFP reporter and compare them to GFP-negative neurons (non-Atf3-expressing) in the ipsilateral and contralateral cortices. Because Atf3 is DLK dependent (Supp Fig 12e) crossing DLK flox mice to the Atf3-Cre line would not produce mice where Atf3-expressing neurons lack DLK.

An alternative to this approach would be to use our layer V Dlk cKO mice in which the Cre+ nuclei are labeled (figure 5c) and compare labeled ipsilateral Dlk cKO neurons to contralateral and/or sham neurons. The problem with this approach is that only a subset of layer V neurons activates Atf3 after injury and, because we know that ipsilateral Atf3-expressing neurons are significantly different from other ipsilateral neurons, we would be unable to conclude whether the neurons recorded from would have been Atf3-expressing in the presence of Dlk. In fact, performing this experiment could give us the false impression that Dlk cKO neurons are similar to contralateral neurons just because we recorded from the wrong cells. Performing this experiment would require recording from 100s of neurons across multiple mice to be confident of the results. Still, we agree with the reviewer that recording from surviving Dlk cKO neurons after an injury would be of great interest, and might be better performed in an injury model in which a distinct and isolated population is consistently injured.”

Rebuttal for final submission

REVIEWERS' COMMENTS

Reviewer #1 (Remarks to the Author):

Most of the concerns raised in the original review have been addressed. The remaining concerns that remain can be addressed with a small caveat section in the discussion.

These include:

- consistent use of contralateral as control when sham injured animals are the standard in the field. This is especially important for mTBI models that can have diffuse injury in the contralateral cortex.

In figure 1, we show that there is no difference between sham and contralateral for our readouts of interest. Therefore we surmise this particular mTBI model does not exhibit diffuse injury for any of the metrics that we investigate in this study.

- Low n number (3 animals for multiple studies, 2 technical replicates for scRNAseq) for some experiments.

We agree with the reviewer and we are transparent about the n number and replicates in the manuscript.

Reviewer #2 (Remarks to the Author):

Overall, the revision addresses many of the concerns expressed by the reviewers, including welcome updated and new figures. However, some issues remain to be clarified before publication.

1) Regarding the choice of focus on ATF3-positive cells (this was raised by the other reviewers too): some of the arguments and data in the rebuttal provides useful context and it would be helpful to incorporate these (perhaps as supp data) into the paper: e.g., Rebuttal page 25: “We were able to find Atf3-expressing cells, but they clustered separately and had no counterpart in the Sham data. This cluster, consisting of approximately 55 cells...).”

While we agree this information is helpful for context, we think the data we provided in the rebuttal is not complete enough to include as a supp figure (females only, n=1 replicate, etc...)

2) The observation that ATF3 induction represents one of potentially a number of injury programs should be highlighted in both the Abstract and the main text, so that the significance is not inadvertently overinterpreted in the literature. Only about 10% of Layer V YFP neurons were found to be ATF3+ at 7 dpi, and at this time point, 15% of YFP neurons are already dead, but not necessarily because of ATF3. This point can be better addressed

in Introduction-Paragraph 3 and certainly, in the Discussion - Limitations section. We also suggest that ATF3+ neurons always be identified as such, to avoid giving the impression that this injury-induced program may be applied more broadly than it can, e.g. “ATF3-Layer V neurons die within 14 days post injury (dpi), while ATF3-Layer II/III neurons persist and retain firing capabilities.” Clarifying the percentage of ATF3 responsive neurons across layers / regions would be helpful.

We do not think that our phrasing overclaims anything about ATF3. We frame the entire manuscript around investigating the ATF3 response in cortical neurons after mTBI. We never claim it is the only injury response.

We have added this sentence to the Discussion-Limitations section:

Of note, our focus on ATF3 induction in this study does not preclude the existence of additional injury programs in cortical neurons after mTBI.

3) In Rebuttal, in response to transient ATF3 expression after injury: “With our Cre-dependent genetic reporter strategy, even transient Atf3 expression should induce production of the reporter.” This is not true for inducible Cre (CreER), where tamoxifen induction and ATF3 expression must coincide for CreER to be effective. Relevant to this: “We employed targeted snRNAseq of Atf3-expressing neurons using an inducible Atf3-IRES-CreER mouse line crossed to the INTACT nuclear envelope protein reporter” (Page 5). Also: “This mouse results from a cross between Atf3-IRES-Cre and a Cre dependent reporter line expressing GFP under control of the neuron-specific Snap25 promoter (Jax 021879).” Thus, both inducible and non-inducible forms of Cre were used in the study, and accordingly, whether Cre or CreER was used in a particular experiment should be stressed when presenting results. This is why the absence of a rigorous description of the ATF3-GFP strain in the original submission was so confusing, as both Cre and CreER mice were used in this study. Also, need to clarify if ATF3-IRES-Cre = ATF3-Cre and if ATF3-IRES-CreER = ATF3-CreER here. For CreER, the timing of Tamoxifen treatment is important when interpreting results.

We agree the descriptions in the initial submission were confusing. We think the current version is clear about ATF3-IRES -Cre versus -CreER.

4) Why does UMAP in new Fig. 2e looks so different from the old one? Annotation with a reference atlas should not change the shape of UMAP.

The UMAP looks different now because in our revision we re-ran the entire analysis according to the code we have deposited on GitHub. Removal or addition of just a single cell can change the way a UMAP looks. This does not in any way alter our conclusions.

Minor:

Line 154: Missing % value for inhibitory neurons.

This has been inserted (9.2%).

Reviewer #3 (Remarks to the Author):

Reviewer #4 (Remarks to the Author):

The authors have thoughtfully and carefully addressed my primary concerns and I do not have additional concerns. This manuscript reveals important neuronal populations specific effects after TBI.

Critique:

Figure S10i-k the image and quantification of *Atf4* is missing.

The reviewer is correct. In the original manuscript we only had $n=2$ for this quantification. We were unable to increase the sample size for *Atf4* quantifications so we opted to remove the figure entirely. The point of this figure was to show ISR being downstream of DLK. We still show this using *Chop* transcript (S11j), so the interpretation does not change. Furthermore, based on current knowledge in the ISR field, *Chop* transcript is expected to be more highly increased than *Atf4* (which is regulated mostly at the protein level).